# Tracing social mechanisms and interregional connections in Early Bronze Age Societies in Lower Austria

Anja Furtwängler[1] ✉, Katharina Rebay-Salisbury [2,3] ✉, Gunnar U. Neumann [1], Fabian Kanz [4], Harald Ringbauer [1,5], Raffaela Angelina Bianco [1], Tanja Schmidt[1], Lena Semerau[1], Rita Radzevičiūtė[1], Rodrigo Barquera [1], Nadin Rohland[6], Kristin Stewardson[6], J. Noah Workman[6,7], Elizabeth Curtis[6,7], Fatma Zalzala[6,7], Kim Callan [6,7], Lora Iliev[6,7], Lijun Qiu[6,7], Olivia Cheronet [8,9], Anna Wagner [8,9], Guillermo Bravo Morante[8,9], Michaela Spannagel[10,11], Maria Teschler-Nicola [8,11], Friederike Novotny[11], Domnika Verdianu [2,3], Ron Pinhasi [8,9], David Reich [5,6,7,12], Johannes Krause [1,13], Philipp W. Stockhammer[1,13,14] & Alissa Mittnik [1,5,13] ✉

In this study, we present the results of archaeogenetic investigations of Early Bronze Age individuals from Lower Austria, specifically associated with the Únětice and Unterwölbling cultural groups. Through analysing newly generated genome-wide data of 129 individuals, we explore the social structure and genetic relationships within and between these communities. Our results reveal a predominantly patrilocal society with non-strict female exogamic practices. Additionally, Identity-by-Descent analysis detects long-distance genetic connections, emphasizing the complex network of interactions in Central Europe during this period. Despite shared social dynamics, notable genetic distinctions emerge between the Únětice and Unterwölbling groups. These insights contribute to our understanding of Bronze Age population interconnections and call for a nuanced interpretation of social dynamics in this historical context.

The transition from the Final Neolithic to the Early Bronze Age in central Europe during the third millennium BCE was marked by significant socio-cultural transformations and population dynamics. The migration of people from the Pontic-Caspian Steppe into Central Europe reshaped the genetics of local populations through a multi-generational process of mobility, interaction, and gradual admixture[1–4]. These interactions occurred alongside ongoing metallurgical developments, whereby bronze technology was appropriated in different speeds and intensities throughout the late 3rd and early 2nd millennium BCE Central Europe[5,6]. As Vandkilde[7,8]

[1]Department of Archaeogenetics, Max Planck Institute for Evolutionary Anthropology, Leipzig, Germany. [2]Austrian Archaeological Institute, Austrian Academy of Sciences, Vienna, Austria. [3]Department of Prehistoric and Historical Archaeology, University of Vienna, Vienna, Austria. [4]Center of Forensic Medicine, Medical University Vienna, Vienna, Austria. [5]Department of Human Evolutionary Biology, Harvard University, Cambridge, MA, USA. [6]Department of Genetics, Harvard Medical School, Boston, MA, USA. [7]Howard Hughes Medical Institute, Harvard Medical School, Boston, MA, USA. [8]Department of Evolutionary Anthropology, University of Vienna, Vienna, Austria. [9]Human Evolution and Archaeological Sciences (HEAS), University of Vienna, Vienna, Austria. [10]Institute for Oriental and European Archaeology, Austrian Academy of Sciences, Vienna, Austria. [11]Department of Anthropology, Natural History Museum Vienna, Vienna, Austria. [12]Broad Institute of MIT and Harvard, Cambridge, MA, USA. [13]Max Planck Harvard Research Center for the Archaeoscience of the Ancient Mediterranean (MHAAM), Leipzig, Germany. [14]Institute for Pre- and Protohistoric Archaeology and Archaeology of the Roman Provinces, Ludwig Maximilian University, Munich, Germany. ✉e-mail: anja_furtwaengler@eva.mpg.de; katharina.rebay-salisbury@oeaw.ac.at; alissa_mittnik@eva.mpg.de

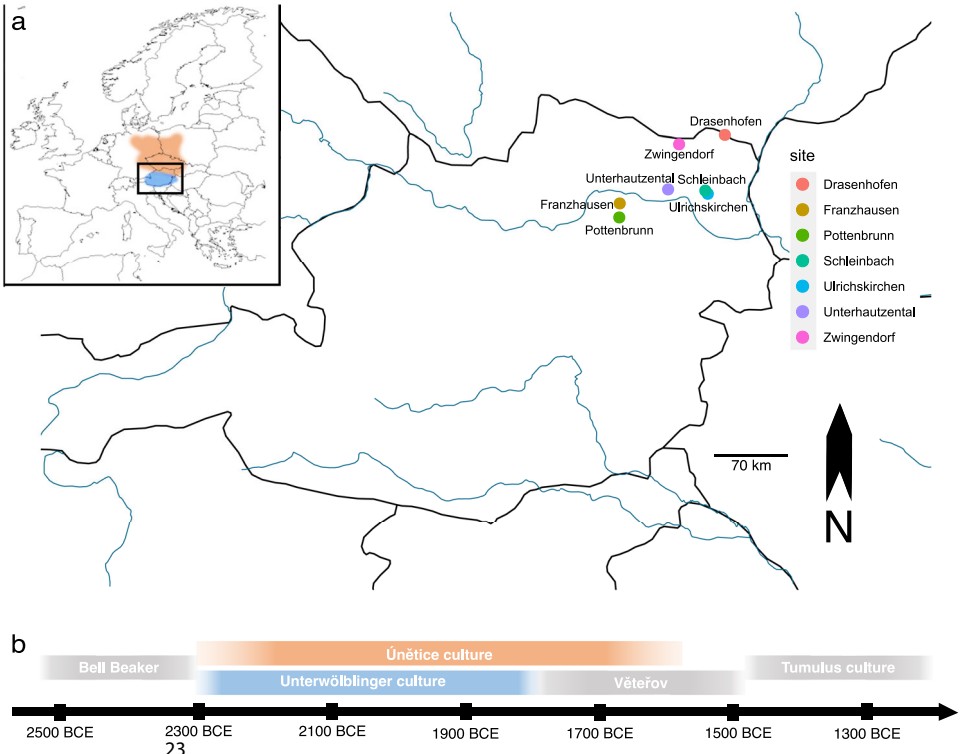

**Fig. 1 | Temporal and geographic distribution of studied Early Bronze Age individuals from Lower Austria. a** Map of Austria showing the locations of sampled sites associated with the Únětice (orange) and Unterwölbling (blue) cultures. Archaeological sites are indicated with coloured circles: dark red – Zwingendorf; bright red – Drasenhofen; orange – Ulrichskirchen; light orange – Schleinbach; dark blue – Franzhausen; light blue – Pottenbrunn; grey-blue – Unterhautzental. **b** Temporal range of the Únětice and Unterwölbling culture in Lower Austria. Basemap from Natural Earth (public domain).

argues, the spread of bronze later played a major role in shaping long-distance connectivity and cultural transmission, which she defined as "Bronzization", acting as a key driver of sociopolitical networks across Eurasia.

Studies across different regions of Central Europe have documented compelling evidence for gradual genetic homogenization among populations during the Early Bronze Age (EBA). These studies reveal striking patterns of genetic similarity and shared ancestry among EBA individuals across large geographic areas, underscoring the extensive genetic interconnectedness[4,9–11]. Moreover, these EBA populations exhibit notable similarities in settlement patterns and subsistence strategies, characterized by homesteads comprising residential and farming structures close to cemeteries (see Supplementary Note 1: Archaeological Information on the sites for more details).

Previous research has provided evidence for patrilocality during the Early Bronze Age and the preceding Late Neolithic/Copper Age, with studies indicating that sons inherited the farmstead while daughters married into other groups[9,11,12]. However, interpretations of prehistoric kinship systems remain an ongoing discussion, and alternative models have been proposed, particularly for earlier periods such as the Neolithic[13,14].

These insights of largely homogenous and patrilocal EBA populations in Central Europe were mainly derived from an interregional perspective. However, the question remains how these dynamics manifested within smaller, more localized key regions.

To address this question, we focused on Lower Austria north of the Eastern Alps, a region with rich archaeological evidence from the Early Bronze Age, covering a discrete area of 80 kilometres in radius (Fig. 1). The area is divided by the river Danube, which gave rise to the formation of two distinct early Bronze Age cultural expressions in immediate vicinity: the Únětice and the Unterwölbling groups, which differ significantly in their burial custom.

The area north of the Danube is part of the Únětice complex, known for its rich archaeological record with items such as the famous Nebra Sky Disc and for social stratification expressed in early 'princely' burials such as those at Leubingen and Helmsdorf in Central Germany[15,16]. In contrast, the archaeological evidence from Lower Austria is more modest[17]. It consists of small farmsteads, often near rivers and bodies of water, with settlement burials in former storage pits, and cemeteries with few graves near the settlements. Some graves are arranged in rows in small plots, and while a single burial per grave is commonplace, instances of double and multiple burials are also present. The predominant burial practice involved positioning both men and women's bodies in a flexed position, typically on their right side, oriented with the head in the south and the gaze directed towards the east. Archaeologically preserved grave goods are modest and comprise bone, shell and bronze jewellery and dress items, as well as ceramic bowls, jugs and vessels placed with the body. Tools and weapons such as daggers are rare. Animal bones and botanical remains from offerings of food and drink may also be present. Our archaeogenetic investigation focused on the archaeologically and anthropologically well-contextualized sites of Drasenhofen[18,19], Zwingendorf[20], Unterhautzenthal[21], Schleinbach[22] and Ulrichskirchen[23] (see Supplementary Note 1: Archaeological Information on the sites for more information).

The Unterwölbling Group south of the Danube and west of the Viennese woods culturally connects to the Early Bronze Age southern German groups[24]. Among the well-known sites from this region are the farmsteads and cemeteries from the Lech river valley, which have been extensively researched[9,25]. Following Bell Beaker traditions, and in contrast to the Lower Austrian Únětice groups, the burial rites were strongly gendered according to binary sex characteristics, even for children[26,27]. Men were typically placed in a flexed position on their left side with their head to the north, while women were usually positioned

on their right side with their head to the south. Both thus faced towards the east. This practice of sex-differentiated burial was common in large parts of Central Europe, for example, in the southern Rhine area, along the Danube, as well as along large parts of the Tisza and Vistula rivers (see 15 **Meller 2021**: 104 for a map), whereas more idiosyncratic practices are reported for the periphery of Europe[28]. It appears that it was important to the Central European Bronze Age people in these communities to adhere to a binary gender division. However, the way individuals were buried in gendered burials does not necessarily reflect how they identified themselves or performed gender during their lifetime.

Significant cemeteries have been discovered, with hundreds, if not thousands, of burials, such as at Gemeinlebarn F[26] and Franzhausen I and II[29,30]. Single burials are typical, and double and multiple burials are exceedingly rare. The depth of graves and the number and quality of grave goods vary according to the social status of the deceased, although this is difficult to determine due to the frequent reopening of graves when grave goods were removed. In this archaeogenetic study, we included individuals from Franzhausen I[26] and Pottenbrunn[31] (see Supplementary Note 1: Archaeological Information on the sites for more information).

By focusing on the microscale perspective, we aim to determine the level of genetic homogeneity across populations separated by geographically restrictive barriers, identify differing social structures within contemporaneous populations in Lower Austria, and enhance our understanding of the intricate genetic, social and cultural interactions within the EBA societies of Central Europe.

## Results
### General sample overview
We screened 183 prehistoric individuals from seven archaeological sites dating between 2300 BCE and 1600 BCE (see Supplementary Data 1, 2 and 7 and Fig. 1A and 1B) and applied a stringent selection process to ensure the quality of preserved ancient human DNA. Specifically, we enriched 126 individuals for 1,233,013 ancestry informative sites using the "1240k capture panel"[32] of the human genome, while 15 samples from the site of Pottenbrunn were captured using TWIST kits. The 1,200,343 (1240k panel) SNPs on chromosomes 1-22 and X are entirely included in the used TWIST panel, but the latter includes additional SNPs on the Y chromosome and across the genome.

Following enrichment, individuals with insufficient coverage (less than 30,000 covered sites on the 1240k panel) or indications of contamination (more than 5% estimated contamination on the mtDNA and/or X chromosome) were excluded from the analysis, resulting in the removal of 12 samples. As a result, we present a newly reported dataset comprising 129 individuals from the Early Bronze Age in Lower Austria (32 South of the Danube and 106 North of the Danube).

### Genetic analysis of EBA groups in Lower Austria
We initially conducted a principal component analysis (PCA) by projecting the ancient individuals from EBA Lower Austria onto the first two axes. This PCA was constructed using a dataset of 551 modern-day West Eurasian individuals from 67 groups. In addition, we included 267 previously published ancient genomes[4,32–38] as reference points and projected them onto the PCA plot alongside the EBA individuals from Lower Austria.

The resulting PCA plot (Fig. 2) revealed distinctive patterns. As expected, all 138 EBA individuals from Lower Austria clustered between Early Farmers from Anatolia (Anatolia_Neolithic), Western Hunter-Gatherers (WHG), and pastoralists from the Pontic Caspian Steppe (Russia_EBA_Yamnaya_Samara) and exhibited close proximity to previously published contemporaneous groups from central Europe[9,10,35,39,40].

The genetic analysis indicates genetic differences between individuals from the areas north and south of the Danube. These differences are evident through the mean values observed on the second principal component (PC2) of the previously described PCA (Figs. 2 and 3A), as well as in the relative proportions of ancestry components modelled with qpAdm (Fig. 3B, Supplementary Data 3). Specifically, the individuals who inhabited the area south of the Danube (carrier of Unterwölbling culture) from Pottenbrunn and Franzhausen I exhibit a higher relative amount of Early Farmer ancestry in comparison to their Steppe-related ancestry. Conversely, the Únětice individuals north of the Danube display the opposite pattern, with a greater proportion of Steppe-related ancestry relative to Early Farmer ancestry, resulting in a ratio of Yamnaya Samara to Anatolia Neolithic of 1.91 for the Únětice and 0.89 for the Unterwölbling samples. These contrasting genetic profiles indicate significant genetic differences between the two populations, emphasizing the distinct ancestral contributions and genetic dynamics that influenced them. Analysis of runs of homozygosity (ROH, Supplementary Note 4: analysis of runs of homozygosity) indicates that both groups had a large and outbreeding population (as seen from Supplementary Fig. 4) with no indication of unions between consanguineous parents.

When examining a set of functional SNPs (Supplementary Data 6), which for example includes lactose persistence in adulthood, there were no frequency differences observed between the two groups.

### Intercultural connections between individuals associated with the Únětice and Unterwölbling culture
The investigation utilizing Identity-by-Descent (IBD) analysis allows the detection of genetic connections up to the 8th degree and even identifies connections that extend beyond that range[41]. The analysis of contemporary groups from different regions reveals that the studied individuals from the two cultures generally have the most genetic connections to the Lech Valley (Fig. 4), specifically during the Middle Bronze Age (MBA). This pattern holds true even when considering only the Únětice culture. The Lech Valley, located in present-day southern Germany, was a significant region during the Bronze Age, known for its archaeological richness and evidence of complex social structures, mobility, and martial practices, making it a key reference point for genetic comparisons.

However, focusing solely on the Unterwölbling individuals from our newly generated sequencing data, the strongest genetic connections are found with the Early Bronze Age (EBA) double burial from Bad Zurzach in Switzerland[4].

IBD analysis furthermore reveals minimal connections between the two cultural groups under study, even though all sites in this study originate from an area with no more than an 80-kilometre radius. Only one subadult individual from the Unterwölbling site Franzhausen (FZH002) displays IBD segments shared with individuals from the Únětice group (Fig. 5, separate networks for males and females in Supplementary Note 5: Reconstructed family trees per site, Supplementary Data 5), albeit beyond the 8th degree of kinship. The individual is an adolescent boy from the triple burial of Franzhausen. He was buried with an adult man and a second adolescent boy, who are identified as father and son, to whom he is not genetically related.

The IBD analysis also reveals sex-based differences in social connectivity within Bronze Age cultures, measured by the degree centrality ($k$), defined as the number of links held by each node. In the Únětice culture, females average degree centrality $\langle k \rangle$ is 2.54, compared to 3.42 in males. While this suggests a potential trend toward greater male connectivity and may thus point to gender-differentiated social dynamics, the observed difference in mean values within the Únětice group is not statistically significant (Mann-Whitney U, $W = 57.5$, $p = 0.265$). However, it remains unclear whether this reflects a true absence of difference or is influenced by the limited statistical power due to the small sample size.

Dividing the number of links within site ($k_w$) by the total number of links for each Únětice individual results in an $k_w/k$ ratio of 0.46 for

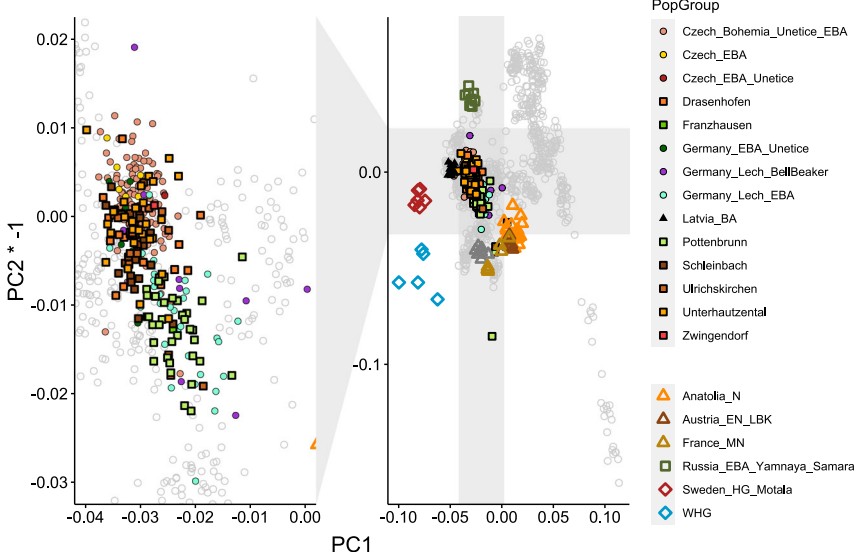

**Fig. 2 | PCA of modern West-Eurasian populations with projected ancient genomes.** Modern West-Eurasian populations are shown as grey circles and ancient genomes, shown as coloured symbols, are projected onto the same PCA space. Source data are provided as a Source Data file.

females and 0.58 for males. This further shows that males are, on average, potentially more connected within their site than females. Conversely, in the Unterwölbling culture the difference between female and male is notably smaller, females average 5 links compared to 5.8 in males. Higher link values in males suggest a tendency towards patrilocality[42].

### Genetic kinship relations within cultures and social structure

By integrating multiple methods, including the examination of mitochondrial DNA (mtDNA) and Y chromosomal haplogroups, we conducted a comprehensive analysis of genetic kinship and successfully reconstructed genetic genealogies within burial grounds associated with the two cultural groups (Fig. 6, Supplementary Data 4). These genetic genealogies extended up to three generations, revealing patterns of genetic relationships. It is important to note that genetic relatedness does not necessarily represent social kinship.

Nevertheless, our observations suggest that it was common practice to bury males in the same cemetery as their parents and offspring, which might indicate a reference to familial connections. In contrast, females were more commonly buried solely with their offspring.

In one genetic genealogy (Supplementary Fig. 5) there are three siblings, two full-siblings with one half-brother who shares the same father, an individual not among those sampled. Because DSH023 and DSH027 share X chromosome similarities (Supplementary Data 4), this may suggest that DSH027's mother and DSH009, the mother of DSH008, share a distant common ancestor. However, the number of shared SNPs on the X chromosome is limited.

Our findings also unveiled two instances in the Únětice group where adult women were buried with their parents or siblings (Supplementary Fig. 4 and Supplementary Fig. 11), which was unexpected based on previous research. In other regions of Central Europe during the Early Bronze Age, evidence of female exogamy was prevalent[9]. Our observation could indicate deviations from practices of exogamy, albeit burial practices might not mirror social behaviour exactly.

Two of these women buried at Drasenhofen also have direct descendents buried in the same burial ground, daughter DSH016 a newborn son, and daughter DSH010 a granddaughter. This might indicate that biological mothers were also involved in these exceptions to female exogamy. These cases of adult women buried alongside close relatives suggest a more nuanced social dynamic, potentially

indicating different forms of familial and social organisation within the populations North and South of the Danube.

## Discussion

The findings from this study present a notable advancement in our understanding of the Early Bronze Age (EBA) communities in the Danube region, as the near-exhaustive sampling of individuals for nuclear DNA from the burial grounds allowed for the construction of a rich and comprehensive dataset. This dense sampling has provided an unprecedented opportunity to gain a detailed snapshot of the people inhabiting this specific area during this prehistoric period.

Despite their geographical proximity, significant genetic differences exist between the Unterwölbling and Únětice groups. The individuals associated with the Unterwölbling culture exhibit a higher genetic ancestry related to the first farmers from Anatolia, in contrast to the higher proportion of steppe-related ancestry observed in the Únětice individuals. This pattern counters the geographical prediction that populations in close proximity are genetically similar and implies that the process of population homogenization in Central Europe during the EBA was driven by culturally shaped interconnectivity and trading networks, and was not a blanket phenomenon. Rather, these nuanced genetic distinctions point to regional diversity in contact between communities, with some groups building deeper ties to specific regions and peoples. Higher mobility between these groups, including by women, is one result of these connections. The genetic similarities on the X chromosome between half-brothers with the same father suggest that their respective mothers were distantly related and may have come from the same region or been part of the same long-standing marriage network.

It is important to note that Identical-by-Descent (IBD) links do not necessarily represent direct migration, but may indicate indirect links through a third, unrepresented region. Nonetheless our finding of strong connections of the Unterwölbling group to Bad Zurzach in Switzerland indicates the presence of networks of contact and communication along the Danube River (supported by IBD analysis). The low number of IBD connections to nearby regions in today's Czech Republic, in contrast, might suggest a preference for specific routes, possibly along major rivers like the Danube, which facilitated broader regional exchange. The relevance of the Danubian network during the EBA has long been recognized archaeologically and was termed the "Danubian Sheet Bronze network" ("Danubischer Blechkreis")[43] and

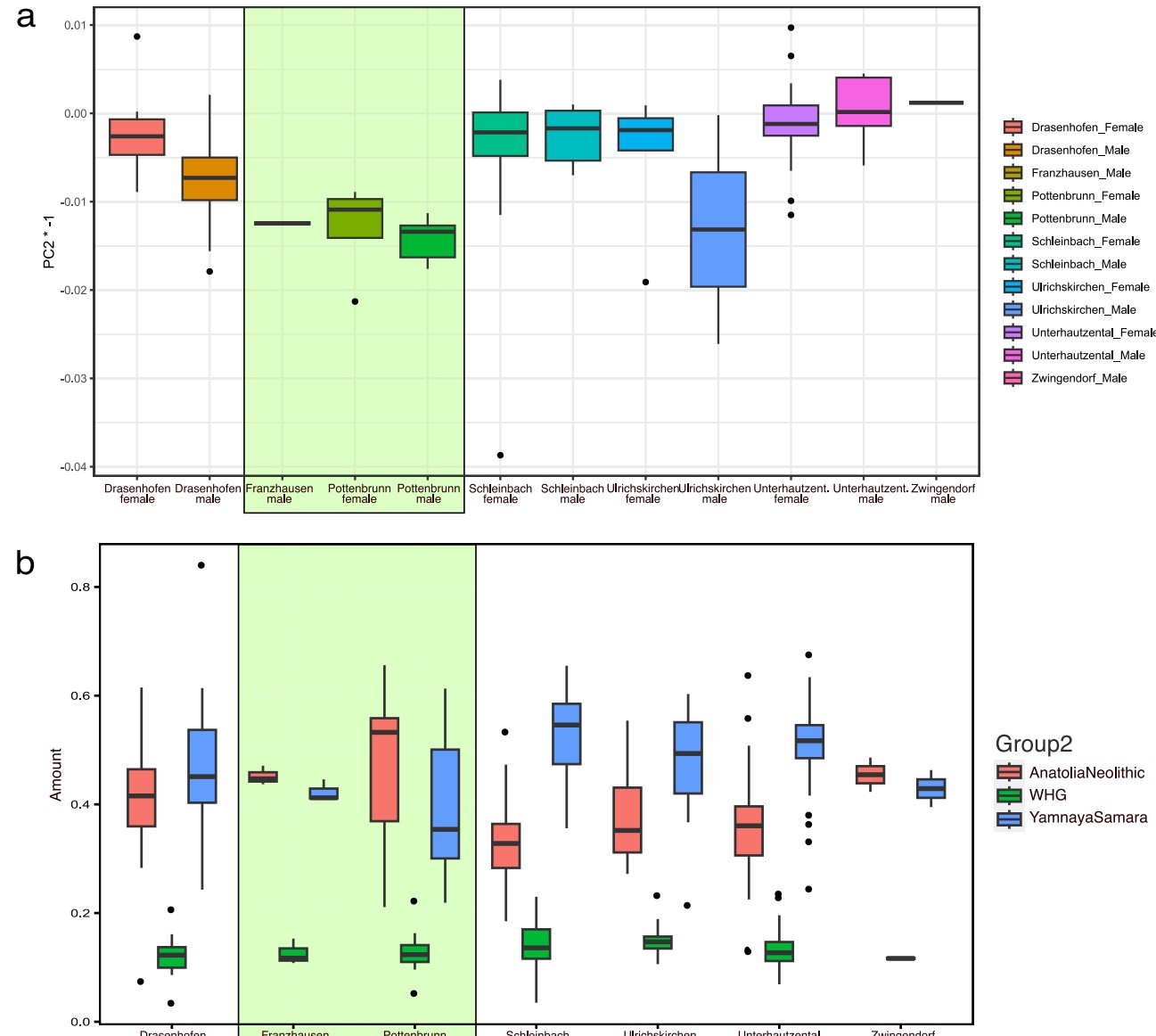

**Fig. 3 | Differences in the mean values of PC2 and the relative amounts of estimated ancestry components for the Únětice and the Unterwölbling associated individuals. a** Mean values of PC2 separated by site and genetic sex, number in brackets indicate sample size. **b** Relative amounts of ancestry components estimated with *qpAdm* for each individual grouped by site, with mean values indicated by lines. Green shading indicates groups associated with the Unterwölbling culture. Groups outside the shaded area belong to the Únětice culture.

the correlation of relatedness in genetics and material culture is relevant to note. However, this relationship existed only during the EBA and there is a marked discontinuity in the Lech Valley between the EBA and the Middle Bronze Age (MBA) with regard to both material culture as well as social systems, which might point to the arrival of new groups in the Lech Valley from ca. 1700 BCE onwards[9]. The genetic link between EBA Lower Austria and MBA Lech Valley is, therefore, unexpected. Thus, our findings might indicate that the population in the MBA Lech Valley likely originated from a group with strong connections to EBA Austria.

Previous studies have consistently identified patrilocality as a prevailing societal pattern in EBA Central Europe. However, this study uncovers instances of women of reproductive age who were buried with their fathers or brothers, along with their own offspring, in the Únětice cultural area north of the Danube. This suggests a link between the less strictly gendered burial practices and less rigid marital rules, where women did not marry into other communities or were returned to their ancestral burial ground after death.

However, alternative explanations should be considered. For instance, such burials might reflect cultural practices where women are interred with natal kin regardless of marital residence, or social circumstances such as marital dissolution, where women returned to their natal households. It is also possible that some women remained unmarried, or that specific rites influenced burial decisions. Additionally, gender identity and the circumstances of death could have shaped these burial choices[14,44,45]. This observation suggests the potential for studies of more nuanced female mobility and varied familial arrangements, ideally involving the study of strontium isotopes, which could enrich the existing interpretative framework and challenge dominant paradigms of patrilocality.

The examination of infants from Unterhautzental reveals the challenges faced by Early Bronze Age children. Three related toddlers from the same family all died at a young age, highlighting the harsh conditions Bronze Age children encountered early in their lives. Traces of injuries and diseases such as meningitis and pleural infection in the skeletal remains of these young individuals[21] further illustrate the

## a

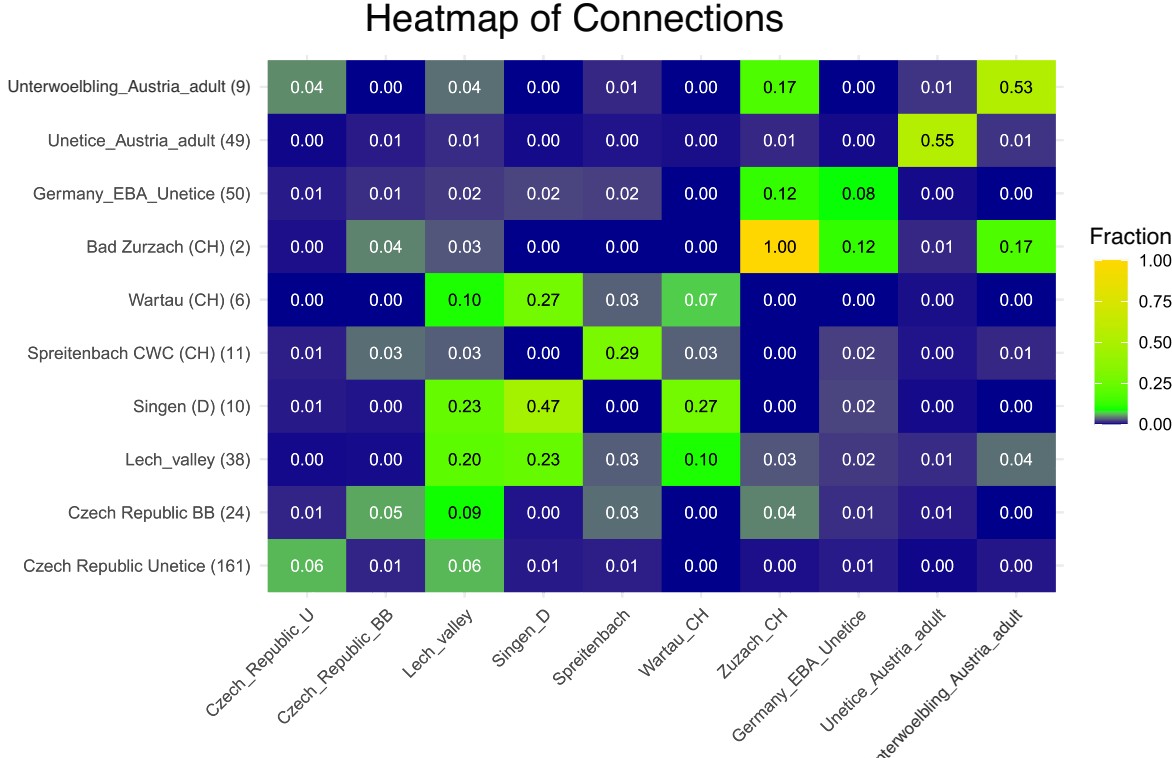

## b

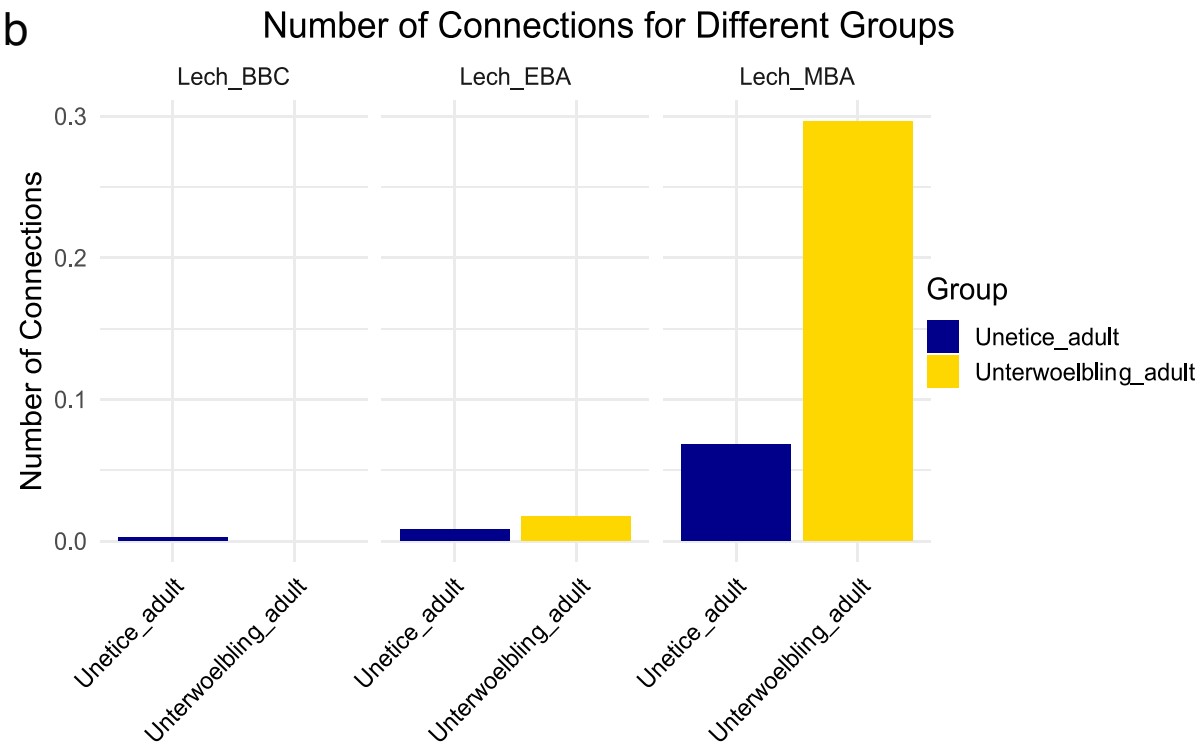

**Fig. 4 | Comparison of IBD connection patterns across different groups.**
**a** Heatmap illustrating the number of IBD connections of segment length 8 cM, 12 cM, 16 cM and 20 cM normalized by the number of possible connections between various groups. The number of possible genetic connections is calculated using the formula $n_1 \times n_2$ for connections between two different groups and ($n \times (n-1)$) / 2 for connections within a single group, where $n_1$ and $n_2$ represent the number of individuals in each group, and n is the number of individuals within a group, following the method suggested by **Ringbaur et al 2024**. The values range from blue (low connection) to yellow (high connection), indicating the strength of the connections. **b** Bar plot showing the normalized number of IBD connections for different groups in the Lech valley, categorized by Únětice and Unterwölbling. The x-axis represents different groups, while the y-axis shows the number of IBD connections. The colours distinguish between the Únětice and Unterwölbling categories. Source data are provided as a Source Data file.

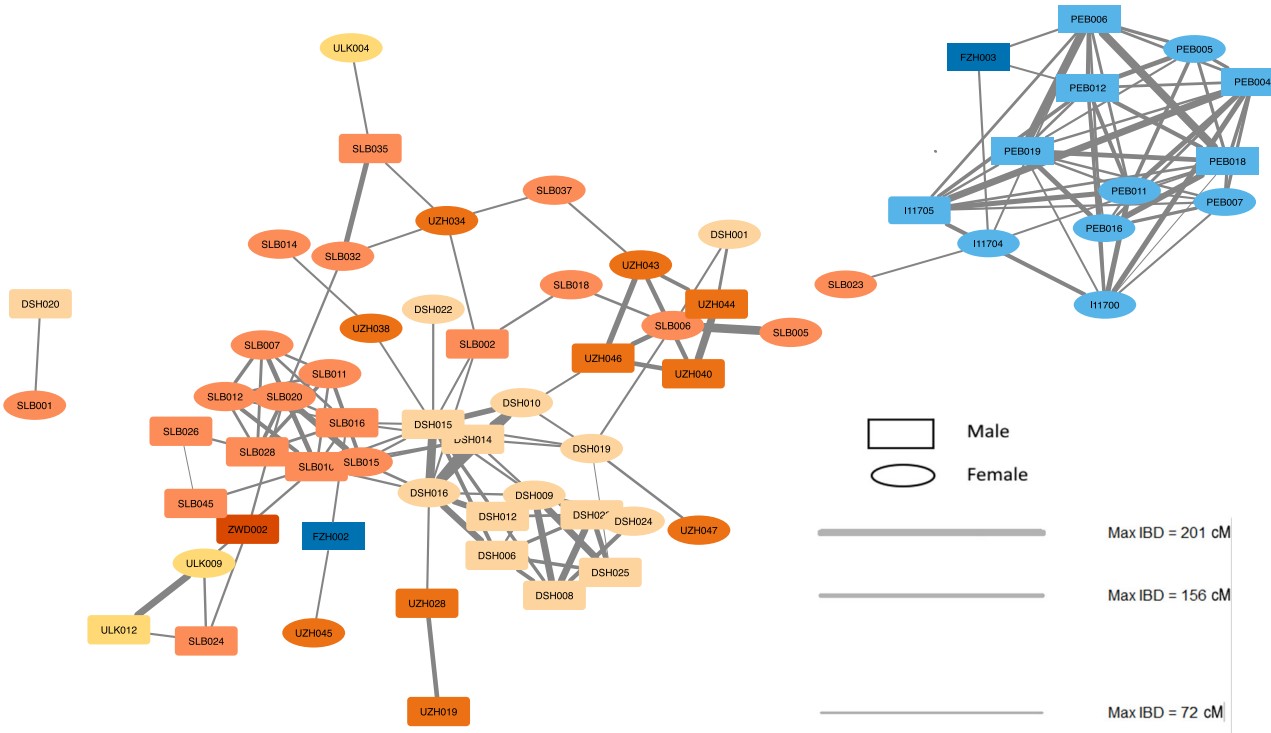

**Fig. 5 | IBD sharing within and between the sites associated with the Únětice and the Unterwölbling culture.** The colours represent the two cultures: blue shades correspond to Unterwölbling, while orange shades correspond to Únětice. Different shades within each colour group indicate distinct sites within the respective culture. The thickness of the lines indicated values of max IBD in centimorgans (cM). Rectangles represent male individuals and ovals represent female individuals.

numerous challenges to health and survival within these communities. These multiple instances of hardships for young children fit well with the previously published case of child murder in Schleinbach, indicating that this was likely not an isolated incident[46].

The extensive dataset, made possible by the dense sampling of individuals from burial grounds, provides a unique perspective on the Early Bronze Age communities in this region. It reveals the complexities of genetic relationships, significant exchange between groups, kinship and social structures. The findings shed light on population dynamics and emphasize the importance of microscale investigations to complement broader regional studies. This invites further exploration into the nuances of human interactions and sociocultural developments during this transformative period in history.

## Methods

Permission to access the human remains analyzed in this study was given by the Department of Anthropology at the Natural History Museum in Vienna, and for the sites of Ulrichskirchen and Drasenhofen, by the Federal Monuments Office of Lower Austria. We collected 188 skeletal elements initially assigned to 183 ancient individuals from seven EBA sites in Lower Austria described in Supplementary Note 1: Archaeological Information on the sites. Some samples have previously been analysed for mitochondrial DNA to assess mother-child relations[47]. All the samples were either teeth or petrous bones. The sampling procedures were conducted within a dedicated aDNA laboratory at MPI-EVA in Jena, by cutting at the cemento-enamel junction and retrieving the inner part of the crown[48,49]. In the case of complete crania from the collection of the Natural History Museum Vienna, bone powder from the petrous part of the temporal bone was collected with a mobile sampling kit at the museum.

Samples were processed using an automated library protocol, which constructs libraries from single-stranded molecules[50]. Each extract produced at least one library that was sequenced at a low depth

on an Illumina HiSeq4000 platform. The raw FastQC files underwent processing through the EAGER pipeline[51] for adaptor removal with AdapterRemoval v2[52], mapping (bwa[53]) against the human reference hs37d5, and PCR duplicate removal (DeDup[54]). The resulting information about library complexity and endogenous DNA percentage was combined with *mapDamage*[55] estimates to assess the preservation of endogenous aDNA. To generate data from a large number of individuals, aDNA enrichment methods, using in-solution hybridization enrichment consisting of approximately 1.2 million ancestry-informative positions (1240k capture[32]) were applied to samples with 0.1% human endogenous DNA or more. Following the 1240k enrichment, the selected libraries were sequenced at standard depth (~20 million reads). Post-1240k capture data was evaluated using EAGER and mapDamage with the same settings. All sequencing data from the same library or multiple libraries from one DNA extract or individual produced with the same protocols were processed equally and merged at the level of bam files. The authentication of aDNA involved three different methods, evaluation of aDNA damage with *mapDamage*[55], and contamination estimated on mtDNA with *schumtzi*[56] and X chromsomes of male individuals with ANGSD[57], on the bam files to estimate modern DNA contamination on ancient samples. Pseudo-haploid genotypes were extracted from the pileups of ss-library bams using pileupcaller (https://github.com/ststschiff/sequenceTools), taking care to filter out bias due to damage.

Sampling of skeletal elements of 15 individuals (Supplementary Data 2) from the site Pottenbrunn was conducted in the dedicated ancient DNA clean rooms of the Pinhasi Lab and DNA was extracted in the ancient DNA clean lab at Harvard Medical School using a method specifically designed to retain short molecules[58,59] and created double-stranded DNA libraries[60,61]. To minimize characteristic base misincorporations that accumulate in ancient DNA strands[62], all libraries were partially treated with Uracil-DNA Glycosylase (UDG). For four libraries, an in-solution hybridization approach to enrich the

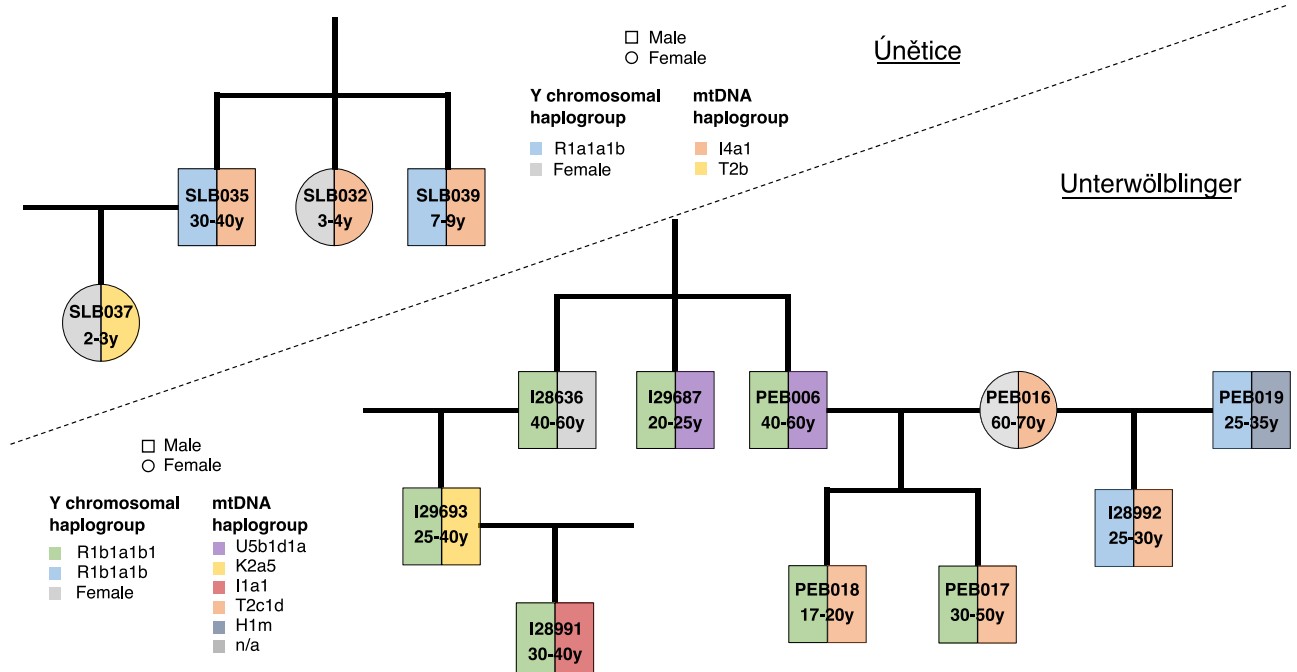

**Fig. 6 | Examples of reconstructed genetic genealogies for the Únětice and Unterwölbling associated sites.** This figure illustrates genetic genealogies from the Early Bronze Age Únětice culture (above the line) and the Unterwölbling culture (below the line). In each diagram, the left half of the circles and squares represent the Y chromosomal haplogroup, while the right half indicates the mtDNA haplogroup. Both cultures exhibit similar patrilocal residence patterns, with women primarily situated within their husband's family units.

libraries for approximately 1.2 million single-nucleotide polymorphisms ('1240k' SNP set)[2,32,63], was employed. For the remaining eleven libraries, the 'Twist' Ancient DNA capture protocol, which targets around 1.35 million SNPs, largely overlapping with the 1240k SNP set, using only a single round of enrichment[58] was applied. Sequencing was performed on an Illumina NextSeq500 instrument using v2 150 cycle kits for 2 × 76 cycles and 2 × 7, or on a HiSeq X10 using v2.5 kits for 2 × 100 cycles and 2 × 7 cycles.

Following data merging and quality control, our final dataset was merged with publicly available genotype datasets of ancient and modern individuals from across Eurasia[4,32–38,64–68]. We performed population structure analysis using PCA with modern West Eurasian populations as a reference and projected ancient individuals onto this PCA to avoid bias introduced by high rates of missing data in aDNA using *smartpca* from the EIGENSOFT package[69].

Relative proportions of ancestry components in the newly sequenced individuals were estimated using *qpAdm* (version: 632) from ADMIXTOOLS[40], https://github.com/DReichLab) using a threshold of 100k SNPs for analysis on an individual level (Supplementary Note 3: outgroup f3 statistics on populations used in IBD analysis) and modern reference individuals (Mbuti, Papuan, Han, and Karitiana) from the Simons Diversity Genome Project dataset[70] and published ancient individuals (Ust Ishim, Villabruna, MA1).

We examined SNPs encoding for biological traits, such as LP, and eye/skin pigmentation, following the list of SNPs used in Supplementary Data 6. For each phenotype-associated locus, we report the number of reads with derived alleles versus the total number of reads covered on this site in Supplementary Data 6, by applying *SAMtools* pileup on BAM files after quality filtering (-q 30 -Q 30).

For the identification of closely related individuals, we employed the READ method[71]. This approach estimates the coefficient of relatedness between two individuals by calculating the rate of mismatching alleles (P0) normalized with the pairwise allele differences among unrelated individuals within the population (α). This normalization

corrects for SNP ascertainment, marker density, genetic drift, and inbreeding.

To detect relatives at a more distant degree, we utilized *lcMLkin*[72] with the options -l phred and/or *ngsRelate*[73,74]. *lcMLkin* leverages a maximum likelihood framework to infer identity by descent (IBD) on low-coverage DNA sequencing data from genotype likelihoods computed with bcftools. The coefficient of relatedness (r) is then calculated as k1/2 + k2, where k1 and k2 represent the probabilities of sharing one or both alleles IBD, respectively. The method can differentiate between parent–offspring (k0 = 0) and siblings (k0 ≥ 0, depending on the recombination rate). To ensure no bias between data generated with the 1240k and the Twist capture, we compared pairwise mismatch rates within data of one capture and between the two captures (Supplementary Note 2: Evaluation of bias from different wetlab methods on genetic kinship analysis). Reconstructed family trees for all sites can be found in Supplementary Note 4: Analysis of runs of homozygosity. This analysis was repeated using only SNPs on the X chromosome, as well as for the entire genome, utilizing the software *ngsRelate*[73].

To estimate Runs of Homozygosity (ROH), we employed the hapROH tool, which is specifically designed for inferring ROH in ancient DNA samples with pseudo-haploid data[75] (Supplementary Note 3: outgroup f3 statistics on populations used in IBD analysis). Genotype likelihoods were called using the MLE function of ATLAS, an ancient-DNA-specific caller (https://bitbucket.org/wegmannlab/atlas/). This was done across approximately 20 million SNPs from the 1000 Genomes Project SNP panel, and these likelihoods were used as input for imputation through GLIMPSE. For imputation, we referenced phased haplotypes from the 1000 Genomes Project phase-3 data and ran GLIMPSE with default settings, using sex-averaged genetic maps from HapMap as recommended[76]. Haplotype IBD analysis was then carried out using ancIBD, a recently developed method that addresses the high error rates in phasing ancient DNA[41].

**Reporting summary**

Further information on research design is available in the Nature Portfolio Reporting Summary linked to this article.

## Data availability

The raw DNA sequences for individuals newly sequenced in this study are deposited in the European Nucleotide Archive under the study accession number PRJEB89777. Source Data for Fig. 3b can be found in Supplementary Data 3. Source Data for Fig. 5 in Supplementary Data 5 and for Fig. 6 in Supplementary Data 4. 1000 Genome Project data, used as the reference panel for the imputation was taken from https://www.internationalgenome.org/data-portal/data-collection/30x-grch38. The reference data used in this study are available in the European Nucleotide Archive under accession code PRJEB11450, PRJEB22652, PRJEB23635, PRJEB30874, PRJEB32466, PRJEB24794, PRJNA608699, PRJEB11364. Human remains analyzed in this study remain under the care of the Department of Anthropology at the Natural History Museum in Vienna (curator of national collections: Karin Wiltschke-Schrotta, karin.wiltschke@nhm.at) and can be found using the accession code NHMW-Anthro-OSTE combined with the inventory number (Inv. Nr.). Individuals from Ulrichskirchen und Drasenhofen remain under the care of the State Collections of Lower Austria (collection manager: Franz Pieler, asparn.urgeschichte@noel.gv.at), accessible through the Federal Monuments Office measure numbers 15128.18.04, 15129.18.04 (Drasenhofen) and 15220.11.01, 15220.11.02 (Ulrichskirchen). Source data are provided with this paper.

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

## Acknowledgements

We thank the staff of the Department of Anthropology at the Natural History Museum in Vienna, in particular Karin Wiltschke-Schrotta, Doris Pany-Kucera and Michaela Spannagl-Steiner, for granting access to the human remains under their curation. We are grateful to Kurt Fiebig, the director of the excavation at Drasenhofen, for the detailed excavation records. Walther Parson and David Reich generously let us access and re-analyze samples and data from previous projects. This study was funded in the framework of the project "The value of mothers to society: responses to motherhood and child rearing practices in prehistoric Europe', which received funding from the European Research Council (ERC) under the European Union's Horizon 2020 research and innovation program (grant agreement No 676828, PI: K. Rebay-Salisbury. This work was supported by National Institutes of Health grant HG012287; by John Templeton Foundation grant 61220; by the Howard Hughes Medical Institute (HHMI), a gift from J.-F. Clin; by the Allen Discovery Center, a Paul G. Allen Frontiers Group advised program.

## Author contributions

K.R.-S. and P.W.S. conceived of the study. A.F., R.A.B., T.S., L.S., R.R., R.B., N.R., K.S., J.N.W., E.C., F.Z., K.C., L.I., L.Q., O.C., A.W., and G.B.M. performed laboratory work. R.P., D.R. and J.K. supervised laboratory work. A.F., G.U.N., A.M. and H.R. and H.R. analyzed data. K.R.-S., F.K., M.S., M.T.-N., F.N., and D.V. performed anthropological assessments. K.R.-S., F.K., M.S., M.T.-N., F.N., R.P. and D.V. assembled and interpreted

archaeological material. A.F. and A.M. designed figures. A.F., A.M., and K.R.-S. wrote the paper with input from all co-authors.

## Funding

## Competing interests
The authors declare no competing interests.
