## [Transparent Peer Review file · Nature Communications]

Tracing social mechanisms and interregional connections in Early Bronze Age Societies in Lower Austria

Corresponding Author: Dr Alissa Mittnik

Version 0:

Reviewer comments:

Reviewer #1

(Remarks to the Author)

Overall comments:

This study screened 198 prehistoric individuals from archaeological sites in Lower Austria, specifically associated with the Únětice and Unterwöbling cultural groups. These individuals date between 2300 and 1600 BCE, corresponding to the Early Bronze Age. Of these, 143 individuals were subjected to capture sequencing. After excluding individuals with insufficient coverage or potential contamination, the study presents a newly reported dataset of 138 individuals from the Early Bronze Age in Lower Austria.

The dataset generated in this study has the potential to be a valuable resource for the research community. However, the analyses conducted on this dataset remain relatively limited, and the conclusions drawn do not offer strong novelty. In particular, the evidence supporting patrilocality appears insufficient. Providing additional statistical testing or comparative analyses would strengthen this interpretation. Expanding the analytical depth and situating the findings more comprehensively—not only within the existing literature on the local region but also in the context of broader European demographic interconnections—would significantly enhance the contribution of this work.

Specific comments:

Lines 120-123: My understanding is that some of the 143 individuals were captured using Twist kits, rather than the 1240k capture panel used in Mathieson et al. (2015). Could you please clarify this point?

Figure 2: It would be helpful to indicate which groups belong to the Únětice or Unterwöbling cultures, as this is not intuitive for readers who may not be familiar with the region.

Line 180: A brief explanation of the Lech Valley would be beneficial, as not all readers will be familiar with its relevance to the study.

Lines 181-183: Could you clarify whether this finding is novel to your study, or if it was already reported in Furtwängler et al. (2022)?

Figure 3: The explanation "The number of possible connections is calculated by $n_1 \times n_2$ between groups and $n(n-1)/2$ within groups" is unclear. Please improve readability of this legend.

Figure 4: The figure is difficult to interpret. The legend mentions only green and orange, but additional colours appear to be present. Improving the figure's resolution and ensuring the colours match the legend would enhance readability. Additionally, what is the unit of max IBD?

Lines 195-198: Which specific individual is being referenced here?

Lines 203-206: If a similar analysis were conducted on another community, what results would be expected? For instance, if there were no sex-biased migration but the sample size was limited, could the observed difference in male-to-female ratios be attributed to sampling bias rather than true male-biased migration? It would be helpful to clarify how this difference is statistically significant.

Lines 262-265: The genetic difference observed between the Early Bronze Age Unterwölbling and Únětice groups is used to support a geographic prediction, but the reasoning behind this statement is unclear. Could you provide further explanation and cite relevant references to support this claim?

Lines 271-272: In addition to IBD, could you provide further evidence supporting this connection, e.g. allele frequency-based methods?

Lines 346-353: Why were two different capture kits used for the samples? How many SNP sites exactly overlap between the two kits?

Lines 354-355: Could you specify which publicly available datasets were used in this analysis? Explicitly listing them would enhance transparency and reproducibility.

Reviewer #2

(Remarks to the Author)

I review this manuscript as an archaeologist knowledgeable about but not expert in the analysis of genetic material from ancient human remains. I restrict my comments to the archaeological framing and interpretation of the genetic data, and not to the laboratory/statistical methods applied to extract and model the genetic data. Since I have recommended some of my own work in my comments below, I waive my anonymity: this review is by Catherine Frieman.

This manuscript presents ca. 140 new full genomes dating to the EBA (2300-1600 BCE) in Austria. It uses these data to present a careful comparison of genetic relationships among groups living in a small geographic area but located on either side of the Danube river and demonstrates genetic differences between these groups. A number of smaller-scale insights are also indicated based on patterns of genetic relatedness within specific cemetery communities and more broadly using IBD methods.

Overall, there is considerable merit in this manuscript. In line with recent norms in aDNA research, this manuscript makes no real argument, but presents a series of new data points and offers (typically quite broad) social interpretations based on these, often in dialogue with other genetic research, but not as deeply with the archaeological literature. The new data are, of course, very welcome. Unsurprisingly, as our knowledge of prehistoric European genetics grows, the complexity and nuance of our models also increases and earlier, simpler models are challenged. In this sense, there is not much that is surprising here, but there is considerable new information. I offer the following comments to help the authors present their data in the most robust and clearest manner, with attentiveness to the sorts of insights offered by genetic data and the wider range of sources which might help them develop the social questions they are keen to address.

At present, the manuscript wobbles frequently between biological and social interpretations – in places within the same sentence – and this slippage indicates the still somewhat uneasy collaboration between archaeologists and geneticists. I have flagged many of these passages and suggested alternative phrasings or questioned the assumptions being made. One particular critique noted several times in the notes that follow is the tendency to present a grand social theory then follow it with a necessary but brief caveat about the limitations of genetic data to allow insight into a given phenomenon. I would suggest to the authors that they consider placing those caveats at the start of their interpretation and then present their interpretation as hypotheses that fit the data to hand, rather than as statements of truth. While it is the genetic data that are novel in this manuscript, for them to be meaningful and important they must be well framed by appropriate archaeological information and social models. This is only sometimes the case within the present manuscript and further work should be focussed on better contextualising and more carefully interpreting the new data.

Major comments:

-The introduction situates the article against some extremely grand claims that are somewhat caricatured, for example that there was a single great migration in the early third millennium (lines 43-46), whereas the lead author's own previous work makes clear even from a genetic standpoint that we are looking at a several-centuries' long and multi-generational process of long-distance movement and interpersonal contact (Furtwängler et al. 2020). The implication that this genetic shift in the 3rd millennium was linked to metal (lines 45-46) is also a bit tenuous. The spread of metallurgy in Europe predates these genetic shifts as well, with copper objects and technology circulating in the 4th millennium BC (Roberts and Frieman 2012, 2015). Bronze appears on the scene much later – from about the last quarter of the 3rd millennium and doesn't become really widespread until second millennium BC. Additionally, it might be worth consulting Vandkilde's (2016, 2019) model of bronze as a framework to discuss bronze as a medium of connection throughout afro-eurasia.

-Lines 55-57 on shared ways of life in EBA central Europe: this should include references to appropriate archaeological literature, or at least reference to the supplementary material where the archaeological context is discussed.

-Lines 58-61: The authors should not present universal patrilocality/patrilineality as a universally agreed upon fact when it is actually a live topic of debate with many facets. Their research group supports one model, other researchers see the data differently. Good citational practice and robust research requires the authors to engage with this critical work seriously and in good faith. They should at least acknowledge the critiques levelled against their interpretative framework, especially Ensor 2021 (focussed on the Neolithic but entirely pertinent), but Frieman et al 2019 also addresses this, though from a slightly different perspective.

-Lines 92-102: I think it's worth being especially careful here about the slippage between sex terminology (male/female) and gender terminology (woman/man). Similarly, you might find Haughton's (2023; Gaydarska et al. 2023) discussion of

ambiguous gendering in Bronze Age British and Irish burials worth reading as comparison to contextualise your data and for the careful terminology he uses. More broadly, Brück's (2009, 2021, 2023; Booth et al 2021; Brück and Frieman 2021) work emphasises the way identities are constructed in the funerary sphere rather than simply reflecting the decedent's own lived reality and this complements what you are trying to do with this passage (see also Frieman 2023).

-Lines 201-204: Similar slippage between sex terms and gender terms. Differences visible in sexed bodies can indicate gender differences but that's inferential rather than implicit.

-Lines 221-226, 231: The authors use the phrase family tree repeatedly in place of the more accurate genetic genealogy (cf. Abel and Frieman 2023). A family tree is a social technology for mapping known relations, an emic tool in anthropological terms. A genetic genealogy is an outsider's tool to map genetic relations, an etic tool, and does not presume a sense of family or connections between any given individual whose genome is included, nor that all individuals included know or recognise these relations to each other. I flag this because the authors note (correctly) at the end of the paragraph (line 225-6) that "It is important to note that genetic relatedness, does not necessarily represent social kinship" but are in some ways forced to include this caveat because their language in previous lines implies otherwise.

-Lines 236-44 & 288-95: the presence of adult women buried with their natal families is given as an exception to female exogamy, which it may be, but also it could very much be part of an exogamic system or any other variety of social system in which (a) adult women who marry out return to natal families when marital partnerships end or are dissolved; (b) adult women who marry out are interred with their natal families under some rites and cosmological systems regardless of the location of their marital partner's households; (c) adult women are not all marriage partners and thus not all are caught up in rules surrounding the funerary treatment of married individuals; (d) the circumstances of death were such that individuals needed to be interred in different ways to wider norms; (e) not all individuals identified genetically as female are gendered 'women' and thus the funerary rites appropriate for them may differ from other genetically female individuals. In short, you are describing variable funerary rites, not living social practices, and should take considerable care in how you compare these to assumed normative structures. For reference to wider literature, see work by Brück (2021, 2023) and Frieman (2023) as well as Frieman et al (2019) as regards the regular mobility of female marriage partners (and their kin) between their marital residence, their natal residence and elsewhere. I would also take care in suggesting that Sr isotopes will necessarily illuminate origins or mobility patterns because of the tendency of Sr signals to become mixed in individuals with high or regular mobility. Unless you have hair or fingernails (as for example Frei et al 2015, 2017), that have very short growth phases, this may not be possible. With petrous bones and molars the local of early childhood (which may also be mixed) and location of burial can be compared, but this does not illuminate mobile itineraries throughout an individual's life.

-Lines 269-70: more slippage between female and women – which is it? Why? What are you emphasising here talking of sex vs gender?

-Line 272: trade networks are not indicated by genetic connections – genetic connections indicate interpersonal contact, including of course regular sexual contact, but not trade. Trade and exchange are (again) social practices whose presence or absence is not illuminated by the movement of genetic material – this should be substantiated with archaeological evidence of trade (which I'm sure exists, but should be cited). If there is no archaeological evidence, then perhaps rephrase to "...indicate the presence of networks of contact and communication along the Danube River."

Lines 284-6: I appreciate the caveat about IBD links, but I think the placement of this phrase at the end of the paragraph is telling. You should put this at the start and then present your interpretation as a hypothesis based on the data rather than presenting your interpretation as fact and then dropping in a caveat at the end.

Lines 301-2: can you clarify the connection between death by privation and death by murder? It is unclear why the former "fits well" with the latter.

Minor style notes:

Introduction: consider moving the first mention of calendar dates into this section as you only say EBA until the first substantive part of the manuscript which does not help your reader situate themselves in time. A chronological table and area map would also be useful for readers not familiar with small regions of Central Europe and their fine-grained periodisation/culture histories. Fig 1 is quite distant from the start of the manuscript and the map is not ideal for situating a reader in space as the inset is very small and the larger map consists of a flat background with unlabelled grey rivers which are unrecognisable to outsiders.

Lines 53-54 – repetition of "large geographic areas"

Fig 2a – I think I'd like some sort of visual indicator like a grey line or dashed line separating the different sites from each other. The data are clearly presented, but since the graph is in a spectrum it is not immediately clear which box plots map individuals from the same site.

Line 186 – typo in caption? "connections od segment length"

Line 195 – typo: "sites" should be "site"

Line 198 – sentence fragment probably should read "One adolescent boy from the triple burial of Franzhausen was buried" rather than "...burial of Franzhausen. He was buried..."

Line 199 – should read "to whom he was not related" rather than "to which"

Fig 4 – The authors should confirm that the red-orange and green are legible for colourblind readers – red-green combos are usually a bad idea – otherwise the figure is exceptionally clear

Lines 227-9: Grammatical/typographic errors make the sentence awkward, consider replacing "there was a common practice among males to be buried in the same cemetery as their parents and offspring, which might indicate a continuation of familial connections" with "it was common practice to bury males in the same cemetery as their parents and offspring, which might indicate a reference to familial connections"

Line 232: "who shares the same father, who was not among the sampled individuals" should be altered to "who shares the same father, an individual not among those sampled"

Line 233-5: I think this is a typo: "the previous partner (DSH027's mother) and DSH009, the mother of DSH008 and DSH027, share a common ancestor" – who is DSH027's mother, the previous partner? DSH009? I think you mean "DSH027's mother and DSH009, the mother of DSH008, share a common ancestor"

Line 240-1: change "indicating that also biological mothers were" to "indicating that biological mothers were also"

-Lines 265-8: This phrasing is unclear “Rather, these nuanced genetic distinctions point towards diverse regional dynamics, where different cultural groups may have chosen specific regions to foster close connections and facilitate the movement of individuals, particularly women.” I think the authors mean something like “Rather, these nuanced genetic distinctions point to regional diversity in contact between communities, with some groups building deeper ties to specific regions and peoples. Higher mobility between these groups, including by women, is one result of these connections.”
Line 276: a better translation would be “Danubian sheet bronze circle” – additionally Kreis does not translate well to English archaeological usage – “network” is probably a less literal but more accurate translation of how -kreis is used in the German literature. Circle would be recognised by European prehistorians as a direct translation from German usage, but it doesn’t really mean anything in English and is not familiar to readers not working regularly with German terminology.
Supplementary information: clear and well presented though with numerous typos and grammatical errors. These files could use a careful proofread for grammar, English usage, and formatting consistency with the main paper (e.g. is it BCE or cal BC?)

References cited

- Abel, S. & C.J. Frieman. 2023. On gene-ealogy: identity, descent, and affiliation in the era of home DNA testing. *Anthropological Science* 131: 15–25.
- Booth, T.J., J. Brück, S. Brace & I. Barnes. 2021. Tales from the Supplementary Information: Ancestry Change in Chalcolithic–Early Bronze Age Britain Was Gradual with Varied Kinship Organization. *Cambridge Archaeological Journal* 31: 379–400. Cambridge Core. <https://www.cambridge.org/core/article/tales-from-the-supplementary-information-ancestry-change-in-chalcolithic-early-bronze-age-britain-was-gradual-with-varied-kinship-organization/5B71BE0F34927E0A7199A6A568DAB3BC>. <https://doi.org/10.1017/S0959774321000019>.
- Brück, J. 2009. Women, death and social change in the British Bronze Age. *Norwegian Archaeology Review* 42: 1–23.
- . 2021. Ancient DNA, kinship and relational identities in Bronze Age Britain. *Antiquity* 95: 228–37. <https://www.cambridge.org/core/journals/antiquity/article/ancient-dna-kinship-and-relational-identities-in-bronze-age-britain/3692A6ADBBB29F4C2ED9B53960DC5992>. <https://doi.org/10.15184/aqy.2020.216>.
- . 2023. Bronze Age relations: genetics, kinship and gender in Britain, in H. Meller, J. Krause, W. Haak & R. Risch (ed.) *Kinship, Sex, and Biological Relatedness: The contribution of archaeogenetics to the understanding of social and biological relations*. Halle, Saale: Propylaeum. <https://books.ub.uni-heidelberg.de/propylaeum/catalog/book/1280/chapter/18011>. <https://doi.org/10.11588/propylaeum.1280>.
- Brück, J. & C.J. Frieman. 2021. Making Kin: The archaeology and genetics of human relationships. *TATuP – Journal for Technology Assessment in Theory and Practice* 30: 47–52. <https://doi.org/10.14512/tatup.30.2.47>.
- Ensor, B.E. 2021. *The not very patrilineal European Neolithic*. Oxford: Archaeopress.
- Frei, K.M. et al. 2015. Tracing the dynamic life story of a Bronze Age Female. *Scientific Reports* 5: 10431. PMC4440039. PMC. <https://doi.org/10.1038/srep10431>.
- . 2017. A matter of months: High precision migration chronology of a Bronze Age female. *PLoS ONE* 12: doi: <https://doi.org/10.1371/journal.pone.0178834>.
- Frieman, C.J. 2023. Kin and connection: Bodies and relations in archaeology and ancient genetics, in H. Meller, J. Krause, W. Haak & R. Risch (ed.) *Kinship, Sex, and Biological Relatedness: The contribution of archaeogenetics to the understanding of social and biological relations*. 15. Mitteldeutscher Archäologentag vom 6. bis 8. Oktober 2022 in Halle (Saale): 43–50 (Tagungen Des Landesmuseums Für Vorgeschichte Halle). <https://doi.org/10.11588/propylaeum.1280.c17994>. <https://doi.org/10.11588/propylaeum.1280.c17994>.
- Frieman, C.J., A. Teather & C. Morgan. 2019. Bodies in motion: Narratives and counter narratives of gendered mobility in European later prehistory. *Norwegian Archaeological Review* 52: 148–69. <https://doi.org/10.1080/00293652.2019.1697355>.
- Furtwängler, A. et al. 2020. Ancient genomes reveal social and genetic structure of Late Neolithic Switzerland. *Nature Communications* 11: 1915. <https://doi.org/10.1038/s41467-020-15560-x>.
- Gaydarska, B. et al. 2023. To Gender or not To Gender? Exploring Gender Variations through Time and Space. *European Journal of Archaeology* 26: 271–98. <https://www.cambridge.org/core/journals/european-journal-of-archaeology/article/to-gender-or-not-to-gender-exploring-gender-variations-through-time-and-space/233DF85AFE29A765D05C8735E681CE8>. <https://doi.org/10.1017/eea.2022.51>.
- Haughton, M. 2023. Gender in Earlier Bronze Age Ireland and Scotland. *European Journal of Archaeology* 26: 19–38. <https://www.cambridge.org/core/journals/european-journal-of-archaeology/article/gender-in-earlier-bronze-age-ireland-and-scotland/75D3ACF3B144F53C606BFD7AA4F1C0A7#>. <https://doi.org/10.1017/eea.2022.29>.
- Roberts, B.W. & C.J. Frieman. 2012. Drawing Boundaries and Building Models: investigating the concept of the ‘Chalcolithic frontier’ in northwest Europe, in M.J. Allen, J. Gardiner & J.A. Sheridan (ed.) *Is there a British Chalcolithic: People, place and polity in the later 3rd millennium*: 27–39. London: Prehistoric Society Research Papers.
- . 2015. Early metallurgy in western and northern Europe, in C. Fowler, J. Harding & D. Hofmann (ed.) *The Oxford handbook of Neolithic Europe*: 711–28. Oxford: Oxford University Press.
- Vandkilde, H. 2016. Bronzization: The Bronze Age as Pre-Modern Globalization. *Præhistorische Zeitschrift* 91: 103–23. <https://doi.org/10.1515/pz-2016-0005>. <https://doi.org/doi:10.1515/pz-2016-0005>.
- . 2019. Bronze Age Beginnings – a Scalar View from the Global Outskirts. *Proceedings of the Prehistoric Society* 85: 1–27. <https://www.cambridge.org/core/journals/proceedings-of-the-prehistoric-society/article/bronze-age-beginnings-a-scalar-view-from-the-global-outskirts/34E9A3EFA006D973825CABC30B93A2A0>. <https://doi.org/10.1017/ppr.2019.7>.

Reviewer #3

(Remarks to the Author)

The authors provide an impressive study of kinship, and social and cultural interactions in Early Bronze Age Austria. The archaeogenetic study is based on 138 new individuals associated with the Únëtica and Unterwöbling cultures, which revealed clear differences between the two groups regarding sociocultural practices and plausible trade networks.

I welcome the local/regional approach and their argument that patterns on interregional scale can only be explained, understood, and questioned, through in-depth analyses of smaller societal units. The author clearly shows that the microscale approach provides evidence of complexity which is not detectable through macroscale analyses. Furthermore, the authors provide a nuanced vocabulary regarding both discussions about identity and about possible differences between biological and social kinship, and contest through the results the ruling idea of a general patrilocal Bronze Age society. The scientific approach provides support for variance in sociocultural practices between societies also within the same cultural tradition. These efforts show that it is of utter importance to continue archaeogenetic research on local and regional scales to make sense of as well as nuance the narratives provided by macroscale approaches.

I argue that this is a very important study for broadening our understanding of Bronze Age social and cultural networks in Austria in particular, but on European level in general. The authors show that geographical barriers, such as the Danube River, can function as separation between populations, but also the opposite. While there were clear differences in ancestry between populations north and south of the Danube, there are clear links in ancestry following the river. This supports the river's function as an important trade route. Furthermore, the results also clearly point out that there are local or regional variance in marital practices, which are only visible through these in-depth analyses.

The study will have significant impact to the field as it both show how geographical boundaries can function as barriers and possibilities to connect as well as, very welcome, nuances the present picture of strict patrilocal communities. This assumption has been contested within traditional archaeology but can now also be shown with the aid of archaeogenetics. The paper is significant since it encourages further bottom-up studies of local or regional scales, and clearly shows the advantages with this kind of approach. By conducting similar studies on new sites in other regions it is possible to add on and gain better understanding of prehistoric sociodynamics and its link to e.g. trade networks. As the authors also state, future studies should preferably also combine the archaeogenetic data with strontium isotope analysis. Laser ablation sampling might provide high resolution data of individual mobility that can deepen the understanding of both sociocultural practices such as exogamy as well as of trade networks and social status.

The material and amount of new data is impressive, and the applied methods are, to my knowledge not being a geneticist, thoroughly described and well adapted to the research questions. The use of IBD analysis opens up for high-resolution understanding of ancient migration, mobility, and sociocultural practices, and has been crucial for gaining the results provided in the paper. The conclusions of the paper are well-founded based on the aims of the paper as well as the presented results.

I think that the article provides important results as well as a nuanced and much needed discussions about their findings as well as links to previously published results on human health from the sites studied. I believe that this article will be a significant contribution to the research community and highly recommend publication! However, I have some minor suggestions that would improve the paper as well as some very minor edits that should be addressed.

Comments and suggested improvements:

Starting around line 78 and continuing to line 102: Description regarding burial practices denotes that males and females were buried in similar, or different manners. Are all individuals in these graves sexed through osteological and/or chemical analyses or are the authors referring to archaeological or mixed gender/sex determination of the graves? If so, I suggest that the terminology should be changed to men and women to acknowledge the distinction between social and biological sex. In line 100-102 the authors change the terminology to men and women and also point out potential complexities regarding gender identity.

Lines 126-127: Dataset of 138 new individuals – it would be helpful to get information about the relation in samples between the two study areas, maybe in brackets.

Lines 162-166: The authors inform about the presence of LP in the individuals; although the information is important and interesting, the statement reads a little abrupt and perhaps a little out of place since research on the frequency of LP in these individuals have not been introduced earlier in the paper. I suggest introducing this question in the paragraph spanning lines 111-115 or omitting this part since it is not later followed up.

Lines 201-209, especially line 207: Here, the number of shared links in male/females are provided. Both cultures exhibit higher male averages. However, the authors state that the Unterwölbling culture conversely have higher male than female averages. I am not sure about how this is conversely, since both cultures seem to exhibit higher link values in males. Either there seems to be a case of the use of wrong word, or, the text needs to be more clearly written.

Lines 236-242: Here the authors state that there are two instances where adult females have been buried at the same burial ground as siblings, parents, and offspring, hence showing exceptions to female exogamy. The authors also provide a description of typical burial practices in the beginning of the paper, where Únětice burials are not gendered. Could there be a possible link between these ungendered burial practices and the less strict marital practices that could be explored in the discussion? Are there local/regional differences in burial practice?

Minor edits:

Line 71: I am not convinced about the use of "archaeological treasures"; I think the wording sounds unscientific and a little colonial. I suggest a change of word.

Lines 75-76: "A few dozen" and "sometimes" are very unprecise values – I think that being more concrete would improve the text as it in current state makes the reader wonder about numbers.

Line 164: A full stop or ; after "...was generally low"...

Line 196: remove "s" in individuals.

Lines 198-200: These sentences read oddly and needs to be revised; The sentence starting with "One adolescent boy" is incomplete.

Line 313: An "l" is missing in "child"

Line 341: An "r" is missing in "Harvard"

Line 374: Identical by descent should be changed to Identity by descent.

Version 1:

Reviewer comments:

Reviewer #1

(Remarks to the Author)

I appreciate the authors' thorough responses to my previous comments and their efforts in revising the manuscript. The revisions have adequately addressed all of the concerns I raised. The clarifications and additional data and analyses significantly improve the clarity and strength of the manuscript. I have no further comments.

Reviewer #2

(Remarks to the Author)

I congratulate the authors on responding well to the reviews. They have answered my questions, clarified their text and made a very strong contribution to the wider European aDNA literature.

An optional edit: i would love to see the rivers labelled (the the very least the Danube) in Fig 1

-Catherine Frieman

Reviewer #3

(Remarks to the Author)

The authors provide an impressive study of kinship, and social and cultural interactions in Early Bronze Age Austria. The archaeogenetic study is based on 138 new individuals associated with the Únětice and Unterwöbling cultures, which revealed clear differences between the two groups regarding sociocultural practices and plausible trade networks.

The paper underscores how microscale analyses can add to the understanding of the complexities of social dynamics in the past which may remain unreveiled through macroscale analyses. I welcome the local/regional study and the in-depth approach to specific sites. The paper provide strong support for biosocial connections between regions, previously unknown, which may aid in future analyses of e.g. trade networks. The paper also challenges an (often) assumed clear patrilocal society of prehistoric Europe and argues for alternative local/regional variants of social practices. These variabilities can only be detected through the type of local in-depth approaches applied in this paper. Furthermore, the results here also ties on to previous bioarchaeological/paleopathological studies conducted on some of the human remains, which deepens the understanding of the demographic pattern in the cemetery. I argue that the paper is highly relevant to the field and related fields.

The revised paper provide sound argumentation and the conclusions and claims are well supported. The authors apply SotA methods which are all sound and well described.

The authors have thouroughly considered and revised upon comments provided in the first review which has strengthened the already well-written paper further.

I recommend publication of this revised paper!

We appreciate the reviewers' careful reading and encouraging and helpful comments. Our replies to the comments are below. Edits to the main text are highlighted there in yellow.

We would like to note that in the revised version we were able to incorporate additional data from several low-coverage individuals and could reconstruct their genetic relation and add them to our genetic genealogies. This affected four individuals who were identified as siblings within existing pedigrees, as well as one individual who was determined to be the mother of three previously analyzed individuals.

REVIEWER COMMENTS

Reviewer #1 (Remarks to the Author):

Overall comments:

This study screened 198 prehistoric individuals from archaeological sites in Lower Austria, specifically associated with the Únětice and Unterwölbling cultural groups. These individuals date between 2300 and 1600 BCE, corresponding to the Early Bronze Age. Of these, 143 individuals were subjected to capture sequencing. After excluding individuals with insufficient coverage or potential contamination, the study presents a newly reported dataset of 138 individuals from the Early Bronze Age in Lower Austria.

The dataset generated in this study has the potential to be a valuable resource for the research community. However, the analyses conducted on this dataset remain relatively limited, and the conclusions drawn do not offer strong novelty. In particular, the evidence supporting patrilocality appears insufficient. Providing additional statistical testing or comparative analyses would strengthen this interpretation. Expanding the analytical depth and situating the findings more comprehensively—not only within the existing literature on the local region but also in the context of broader European demographic interconnections—would significantly enhance the contribution of this work.

Specific comments:

Lines 120-123: My understanding is that some of the 143 individuals were captured using Twist kits, rather than the 1240k capture panel used in Mathieson et al. (2015). Could you please clarify this point? We added the information that 11 of this samples were capture using TWIST kits.

Figure 2: It would be helpful to indicate which groups belong to the Únětice or Unterwölbling cultures, as this is not intuitive for readers who may not be familiar with the region.

We added green shades indicating Unterwölbling culture and added the following to the figure caption: Green shading indicates groups associated with the Unterwölbling culture. Groups outside the shaded area belong to the Únětice culture.

Line 180: A brief explanation of the Lech Valley would be beneficial, as not all readers will be familiar with its relevance to the study.

We added this sentence: The Lech Valley, located in present-day southern Germany, was a significant region during the Bronze Age, known for its archaeological richness and evidence of complex social structures, mobility, and martial practices, making it a key reference point for genetic comparisons.

Lines 181-183: Could you clarify whether this finding is novel to your study, or if it was already reported in Furtwängler et al. (2022)?

In order to make this more clear, we changed the sentence to this: However, focusing solely on the Unterwölbling individuals from our newly generated sequencing data, the strongest genetic connections are found with the Early Bronze Age (EBA) double burial from Bad Zurzach in Switzerland (Furtwängler et al. 2022).

Figure 3: The explanation "The number of possible connections is calculated by $n_1 \times n_2$ between groups and $n(n-1)/2$ within groups" is unclear. Please improve readability of this legend.

We added this explanation: The number of possible genetic connections is calculated using the formula $n_1 \times n_2$ for connections between two different groups and $(n \times (n - 1)) / 2$ for connections within a single group, where n_1 and n_2 represent the number of individuals in each group, and n is the number of individuals within a group.

Figure 4: The figure is difficult to interpret. The legend mentions only green and orange, but additional colours appear to be present. Improving the figure's resolution and ensuring the colours match the legend would enhance readability. Additionally, what is the unit of max IBD?

We changed the figure description to The colors represent the two cultures: green shades correspond to Unterwölbling, while orange shades correspond to Únětice. Different shades within each color group indicate distinct sites within the respective culture. The thickness of the lines indicated values of max IBD in centimorgans (cM). Rectangles represent male individuals and ovals represent female individuals.

Lines 195-198: Which specific individual is being referenced here?

We added the ID of the corresponding individual to the text.

Lines 203-206: If a similar analysis were conducted on another community, what results would be expected? For instance, if there were no sex-biased migration but the sample size was limited, could the observed difference in male-to-female ratios be attributed to sampling bias rather than true male-biased migration? It would be helpful to clarify how this difference is statistically significant.

Sample sizes in ancient DNA studies are often low, which can impact statistical power and make significance testing less reliable. Given this limitation, we primarily relied on descriptive statistics to characterize the observed patterns in the data. However, to formally assess the difference, we performed a Mann-Whitney U test, which yielded a non-significant p-value. We added the following paragraph to the result section: While this suggests a potential trend toward greater male connectivity, the small sample size limited the ability to detect statistically significant differences. The Mann-Whitney U test did not reach significance ($W = 57.5$, $p = 0.265$).

Lines 262-265: The genetic difference observed between the Early Bronze Age Unterwölbling and Únětice groups is used to support a geographic prediction, but the reasoning behind this statement is unclear. Could you provide further explanation and cite relevant references to support this claim?

Yes, this was phrased unclear, we added more information to the paragraph to make it more clear: This pattern counters the geographical prediction that populations in close proximity are genetically similar...

Lines 271-272: In addition to IBD, could you provide further evidence supporting this connection, e.g. allele frequency-based methods?

In addition to the IBD analysis, we performed pairwise f_3 -statistics to further investigate genetic affinities to be now found in SI note 3. While both methods highlight a connection with Bad Zurzach, they assess different aspects of genetic relatedness. IBD primarily reflects recent shared ancestry, whereas f_3 -statistics capture broader allele frequency similarities due to common ancestry or genetic drift. The strong connection observed in both analyses reinforces the genetic link between these populations from complementary perspectives.

Lines 346-353: Why were two different capture kits used for the samples? How many SNP sites exactly overlap between the two kits?

The two different panels were used because a part of the samples was processed in different labs where different capture were implemented: in Boston (Twist) and Leipzig (1240k). The 1,200,343 1240k panel SNPs on chromosomes 1-22 and X are entirely included in the used TWIST panel, but the latter includes additional SNPs on the Y chromosome and across the genome (Rohland N, Mallick S, Mah M, Maier R, Patterson N, Reich D. Three assays for in-solution enrichment of ancient human DNA at more than a million SNPs. *Genome Res.* 2022;32(11-12):2068-2078. doi:10.1101/gr.276728.122).

Lines 354-355: Could you specify which publicly available datasets were used in this analysis? Explicitly listing them would enhance transparency and reproducibility.

They are already listed in line 132 and are added again to the method section: 551 modern-day West Eurasian individuals from 67 groups and published ancient genomes (Mathieson et al. 2015, Feldman et al 2019, Mathieson et al 2018, Olalde et al. 2018, Olalde et al. 2019, Furtwängler et al. 2020, Narasimhan et al. 2019, Jones et al. 2015)

Reviewer #2 (Remarks to the Author):

I review this manuscript as an archaeologist knowledgeable about but not expert in the analysis of genetic material from ancient human remains. I restrict my comments to the archaeological framing and interpretation of the genetic data, and not to the laboratory/statistical methods applied to extract and model the genetic data. Since I have recommended some of my own work in my comments below, I waive my anonymity: this review is by Catherine Frieman.

This manuscript presents ca. 140 new full genomes dating to the EBA (2300-1600 BCE) in Austria. It uses these data to present a careful comparison of genetic relationships among groups living in a small geographic area but located on either side of the Danube river and demonstrates genetic differences between these groups. A number of smaller-scale insights are also indicated based on patterns of genetic relatedness within specific cemetery communities and more broadly using IBD methods.

Overall, there is considerable merit in this manuscript. In line with recent norms in aDNA research, this manuscript makes no real argument, but presents a series of new data points and offers (typically quite broad) social interpretations based on these, often in dialogue with other genetic research, but not as deeply with the archaeological literature. The new data are, of course, very welcome. Unsurprisingly, as our knowledge of prehistoric European genetics grows, the complexity and nuance of our models also increases and earlier, simpler models are challenged. In this sense, there is not much that is surprising here, but there is considerable new information. I offer the following comments to help the authors present their data in the most robust and clearest manner, with attentiveness to the sorts of insights offered by genetic data and the wider range of sources

which might help them develop the social questions they are keen to address.

At present, the manuscript wobbles frequently between biological and social interpretations – in places within the same sentence – and this slippage indicates the still somewhat uneasy collaboration between archaeologists and geneticists. I have flagged many of these passages and suggested alternative phrasings or questioned the assumptions being made. One particular critique noted several times in the notes that follow is the tendency to present a grand social theory then follow it with a necessary but brief caveat about the limitations of genetic data to allow insight into a given phenomenon. I would suggest to the authors that they consider placing those caveats at the start of their interpretation and then present their interpretation as hypotheses that fit the data to hand, rather than as statements of truth. While it is the genetic data that are novel in this manuscript, for them to be meaningful and important they must be well framed by appropriate archaeological information and social models. This is only sometimes the case within the present manuscript and further work should be focussed on better contextualising and more carefully interpreting the new data.

Major comments:

-The introduction situates the article against some extremely grand claims that are somewhat caricatured, for example that there was a single great migration in the early third millennium (lines 43-46), whereas the lead author's own previous work makes clear even from a genetic standpoint that we are looking at a several-centuries' long and multi-generational process of long-distance movement and interpersonal contact (Furtwängler et al. 2020). The implication that this genetic shift in the 3rd millennium was linked to metal (lines 45-46) is also a bit tenuous. The spread of metallurgy in Europe predates these genetic shifts as well, with copper objects and technology circulating in the 4th millennium BC (Roberts and Frieman 2012, 2015). Bronze appears on the scene much later – from about the last quarter of the 3rd millennium and doesn't become really widespread until second millennium BC. Additionally, it might be worth consulting Vandkilde's (2016, 2019) model of bronzization as a framework to discuss bronze as a medium of connection throughout afro-eurasia. We changed the wording in lines 43-51 to emphasize a long-term process and avoid misleading causal attribution between a genetic shift and metal: The migration of people from the Pontic-Caspian Steppe into Central Europe reshaped the genetics of local populations through a multi-generational process of mobility, interaction, and gradual admixture.

-Lines 55-57 on shared ways of life in EBA central Europe: this should include references to appropriate archaeological literature, or at least reference to the supplementary material where the archaeological context is discussed.

We referenced SI Note 1 with the detailed archaeological description

-Lines 58-61: The authors should not present universal patrilocality/patrilineality as a universally agreed upon fact when it is actually a live topic of debate with many facets. Their research group supports one model, other researchers see the data differently. Good citational practice and robust research requires the authors to engage with this critical work seriously and in good faith. They should at least acknowledge the critiques levelled against their interpretative framework, especially Ensor 2021 (focussed on the Neolithic but entirely pertinent), but Frieman et al 2019 also addresses this, though from a slightly different perspective.

We changed the wording of the paragraph as follows: Previous research has provided evidence for patrilocality during the Early Bronze Age and the preceding Copper Age, with studies indicating that sons inherited the family home while females married into other groups (Mittnik et al. 2019, Sjögren et al. 2020, Penske et al. 2024). However, interpretations of prehistoric kinship systems remain an

ongoing discussion, and alternative models have been proposed, particularly for earlier periods such as the Neolithic (Ensor 2021, Frieman et al. 2019).

-Lines 92-102: I think it's worth being especially careful here about the slippage between sex terminology (male/female) and gender terminology (woman/man). Similarly, you might find Haughton's (2023; Gaydarska et al. 2023) discussion of ambiguous gendering in Bronze Age British and Irish burials worth reading as comparison to contextualise your data and for the careful terminology he uses. More broadly, Brück's (2009, 2021, 2023; Booth et al 2021; Brück and Frieman 2021) work emphasises the way identities are constructed in the funerary sphere rather than simply reflecting the decedent's own lived reality and this complements what you are trying to do with this passage (see also Frieman 2023).

Thank you for highlighting the critical distinction and relevant literature. We adapted the wording of this paragraph to highlight the complexity of gender performance and variation seen across Europe.

-Lines 201-204: Similar slippage between sex terms and gender terms. Differences visible in sexed bodies can indicate gender differences but that's inferential rather than implicit.

We changed the wording and highlighted the possible inference of gender difference.

-Lines 221-226, 231: The authors use the phrase family tree repeatedly in place of the more accurate genetic genealogy (cf. Abel and Frieman 2023). A family tree is a social technology for mapping known relations, an emic tool in anthropological terms. A genetic genealogy is an outsider's tool to map genetic relations, an etic tool, and does not presume a sense of family or connections between any given individual whose genome is included, nor that all individuals included know or recognise these relations to each other. I flag this because the authors note (correctly) at the end of the paragraph (line 225-6) that "It is important to note that genetic relatedness, does not necessarily represent social kinship" but are in some ways forced to include this caveat because their language in previous lines implies otherwise.

We changed the wording from family tree to genetic genealogy.

-Lines 236-44 & 288-95: the presence of adult women buried with their natal families is given as an exception to female exogamy, which it may be, but also it could very much be part of an exogamic system or any other variety of social system in which (a) adult women who marry out return to natal families when marital partnerships end or are dissolved; (b) adult women who marry out are interred with their natal families under some rites and cosmological systems regardless of the location of their marital partner's households; (c) adult women are not all marriage partners and thus not all are caught up in rules surrounding the funerary treatment of married individuals; (d) the circumstances of death were such that individuals needed to be interred in different ways to wider norms; (e) not all individuals identified genetically as female are gendered 'women' and thus the funerary rites appropriate for them may differ from other genetically female individuals. In short, you are describing variable funerary rites, not living social practices, and should take considerable care in how you compare these to assumed normative structures. For reference to wider literature, see work by Brück (2021, 2023) and Frieman (2023) as well as Frieman et al (2019) as regards the regular mobility of female marriage partners (and their kin) between their marital residence, their natal residence and elsewhere. I would also take care in suggesting that Sr isotopes will necessarily illuminate origins or mobility patterns because of the tendency of Sr signals to become mixed in individuals with high or regular mobility. Unless you have hair or fingernails (as for example Frei et al 2015, 2017), that have very short growth phases, this may not be possible. With petrous bones and molars the local of early childhood (which may also be mixed) and location of burial can be

compared, but this does not illuminate mobile itineraries throughout an individual's life.

We pointed out that burial practices might not reflect social practices and added the considerations about alternative scenarios in the discussion: "However, alternative explanations should be considered. For instance, such burials might reflect cultural practices where women are interred with natal kin regardless of marital residence, or social circumstances such as marital dissolution, where women returned to their natal households. It is also possible that some women remained unmarried, or that specific rites influenced burial decisions. Additionally, gender identity and the circumstances of death could have shaped these burial choices (Brück 2021, 2023; Frieman 2023). This observation suggests the potential for studies of more nuanced female mobility and varied familial arrangements, ideally involving the study of strontium isotopes, which could enrich the existing interpretative framework and challenge dominant paradigms of patrilocality."

-Lines 269-70: more slippage between female and women – which is it? Why? What are you emphasising here talking of sex vs gender?

We changed the wording to women.

-Line 272: trade networks are not indicated by genetic connections – genetic connections indicate interpersonal contact, including of course regular sexual contact, but not trade. Trade and exchange are (again) social practices whose presence or absence is not illuminated by the movement of genetic material – this should be substantiated with archaeological evidence of trade (which I'm sure exists, but should be cited). If there is no archaeological evidence, then perhaps rephrase to "...indicate the presence of networks of contact and communication along the Danube River."

Thank you. We changed the wording as suggested.

-Lines 284-6: I appreciate the caveat about IBD links, but I think the placement of this phrase at the end of the paragraph is telling. You should put this at the start and then present your interpretation as a hypothesis based on the data rather than presenting your interpretation as fact and then dropping in a caveat at the end.

We started the paragraph now with the limitations indicating that we are phrasing a hypothesis.

-Lines 301-2: can you clarify the connection between death by privation and death by murder? It is unclear why the former "fits well" with the latter.

We changed it to this: "These multiple instances of hardships for young children fit well with the previously published case of child murder in Schleinbach, indicating that this was likely not an isolated incident."

Minor style notes:

Introduction: consider moving the first mention of calendar dates into this section as you only say EBA until the first substantive part of the manuscript which does not help your reader situate themselves in time. A chronological table and area map would also be useful for readers not familiar with small regions of Central Europe and their fine-grained periodisation/culture histories. Fig 1 is quite distant from the start of the manuscript and the map is not ideal for situating a reader in space as the inset is very small and the larger map consists of a flat background with unlabelled grey rivers which are unrecognisable to outsiders.

We split Figure 1 in two and place the map and the time scale in the introduction.

Lines 53-54 – repetition of "large geographic areas"

Yes, we removed it one time.

Fig 2a – I think I'd like some sort of visual indicator like a grey line or dashed line separating the

different sites from each other. The data are clearly presented, but since the graph is in a spectrum it is not immediately clear which box plots map individuals from the same site.

We added shaded areas to better separate the displayed data of the two culture/areas under investigation.

Line 186 – typo in caption? “connections od segment length”

Yes, should be of segment length.

Line 195 – typo: “sites” should be “site”

Yes, should have been site.

Line 198 – sentence fragment probably should read “One adolescent boy from the triple burial of Franzhausen was buried” rather than “...burial of Franzhausen. He was buried...”

We rephrased the paragraph.

Line 199 – should read “to whom he was not related” rather than “to which”

Yes, we changed it to whom.

Fig 4 – The authors should confirm that the red-orange and green are legible for colourblind readers – red-green combos are usually a bad idea – otherwise the figure is exceptionally clear

We changed the green to blue. The groups are now better distinguishable.

Lines 227-9: Grammatical/typographic errors make the sentence awkward, consider replacing “there was a common practice among males to be buried in the same cemetery as their parents and offspring, which might indicate a continuation of familial connections” with “it was common practice to bury males in the same cemetery as their parents and offspring, which might indicate a reference to familial connections”

We replaced the sentence as suggested.

Line 232: “who shares the same father, who was not among the sampled individuals” should be altered to “who shares the same father, an individual not among those sampled”

We changed the sentence as suggested.

Line 233-5: I think this is a typo: “the previous partner (DSH027’s mother) and DSH009, the mother of DSH008 and DSH027, share a common ancestor” – who is DSH027’s mother, the previous partner? DSH009? I think you mean “DSH027’s mother and DSH009, the mother of DSH008, share a common ancestor”

We changed the sentence as suggested.

Line 240-1: change “indicating that also biological mothers were” to “indicating that biological mothers were also”

We changed the sentence accordingly.

-Lines 265-8: This phasing is unclear “Rather, these nuanced genetic distinctions point towards diverse regional dynamics, where different cultural groups may have chosen specific regions to foster close connections and facilitate the movement of individuals, particularly women.” I think the authors mean something like “Rather, these nuanced genetic distinctions point to regional diversity in contact between communities, with some groups building deeper ties to specific regions and peoples. Higher mobility between these groups, including by women, is one result of these connections.”

We changed the sentence accordingly.

Line 276: a better translation would be “Danubian sheet bronze circle” – additionally Kreis does not

translate well to English archaeological usage – “network” is probably a less literal but more accurate translation of how -kreis is used in the German literature. Circle would be recognised by European prehistorians as a direct translation from German usage, but it doesn't really mean anything in English and is not familiar to readers not working regularly with German terminology.

We changed it to network.

Supplementary information: clear and well presented though with numerous typos and grammatical errors. These files could use a careful proofread for grammar, English usage, and formatting consistency with the main paper (e.g. is it BCE or cal BC?)

References cited

- Abel, S. & C.J. Frieman. 2023. On gene-ealogy: identity, descent, and affiliation in the era of home DNA testing. *Anthropological Science* 131: 15–25.
- Booth, T.J., J. Brück, S. Brace & I. Barnes. 2021. Tales from the Supplementary Information: Ancestry Change in Chalcolithic–Early Bronze Age Britain Was Gradual with Varied Kinship Organization. *Cambridge Archaeological Journal* 31: 379–400. Cambridge Core. <https://www.cambridge.org/core/article/tales-from-the-supplementary-information-ancestry-change-in-chalcolithicearly-bronze-age-britain-was-gradual-with-varied-kinship-organization/5B71BE0F34927E0A7199A6A568DAB3BC>. <https://doi.org/10.1017/S0959774321000019>.
- Brück, J. 2009. Women, death and social change in the British Bronze Age. *Norwegian Archaeology Review* 42: 1–23.
- . 2021. Ancient DNA, kinship and relational identities in Bronze Age Britain. *Antiquity* 95: 228–37. <https://www.cambridge.org/core/journals/antiquity/article/ancient-dna-kinship-and-relational-identities-in-bronze-age-britain/3692A6ADBBB29F4C2ED9B53960DC5992>. <https://doi.org/10.15184/aqy.2020.216>.
- . 2023. Bronze Age relations: genetics, kinship and gender in Britain, in H. Meller, J. Krause, W. Haak & R. Risch (ed.) *Kinship, Sex, and Biological Relatedness: The contribution of archaeogenetics to the understanding of social and biological relations*. Halle, Saale: Propylaeum. <https://books.ub.uni-heidelberg.de/propylaeum/catalog/book/1280/chapter/18011>. <https://doi.org/10.11588/propylaeum.1280>.
- Brück, J. & C.J. Frieman. 2021. Making Kin: The archaeology and genetics of human relationships. *TATuP – Journal for Technology Assessment in Theory and Practice* 30: 47–52. <https://doi.org/10.14512/tatup.30.2.47>.
- Ensor, B.E. 2021. *The not very patrilineal European Neolithic*. Oxford: Archaeopress.
- Frei, K.M. et al. 2015. Tracing the dynamic life story of a Bronze Age Female. *Scientific Reports* 5: 10431. PMC4440039. PMC. <https://doi.org/10.1038/srep10431>.
- . 2017. A matter of months: High precision migration chronology of a Bronze Age female. *PLoS ONE* 12: doi: <https://doi.org/10.1371/journal.pone.0178834>.
- Frieman, C.J. 2023. Kin and connection: Bodies and relations in archaeology and ancient genetics, in H. Meller, J. Krause, W. Haak & R. Risch (ed.) *Kinship, Sex, and Biological Relatedness : The contribution of archaeogenetics to the understanding of social and biological relations*. 15. *Mitteldeutscher Archäologentag vom 6. bis 8. Oktober 2022 in Halle (Saale): 43–50 (Tagungen Des Landesmuseums Für Vorgeschichte Halle)*. <https://doi.org/10.11588/propylaeum.1280.c17994>. <https://doi.org/10.11588/propylaeum.1280.c17994>.
- Frieman, C.J., A. Teather & C. Morgan. 2019. Bodies in motion: Narratives and counter narratives of gendered mobility in European later prehistory. *Norwegian Archaeological Review* 52: 148–69. <https://doi.org/10.1080/00293652.2019.1697355>.
- Furtwängler, A. et al. 2020. Ancient genomes reveal social and genetic structure of Late Neolithic

- Switzerland. *Nature Communications* 11: 1915. <https://doi.org/10.1038/s41467-020-15560-x>.
- Gaydarska, B. et al. 2023. To Gender or not To Gender? Exploring Gender Variations through Time and Space. *European Journal of Archaeology* 26: 271–98. <https://www.cambridge.org/core/journals/european-journal-of-archaeology/article/to-gender-or-not-to-gender-exploring-gender-variations-through-time-and-space/233DF85AFAEA29A765D05C8735E681CE8>. <https://doi.org/10.1017/eea.2022.51>.
- Haughton, M. 2023. Gender in Earlier Bronze Age Ireland and Scotland. *European Journal of Archaeology* 26: 19–38. <https://www.cambridge.org/core/journals/european-journal-of-archaeology/article/gender-in-earlier-bronze-age-ireland-and-scotland/75D3ACF3B144F53C606BFD7AA4F1C0A7#>. <https://doi.org/10.1017/eea.2022.29>.
- Roberts, B.W. & C.J. Frieman. 2012. Drawing Boundaries and Building Models: investigating the concept of the ‘Chalcolithic frontier’ in northwest Europe, in M.J. Allen, J. Gardiner & J.A. Sheridan (ed.) *Is there a British Chalcolithic: People, place and polity in the later 3rd millennium*: 27–39. London: Prehistoric Society Research Papers.
- . 2015. Early metallurgy in western and northern Europe, in C. Fowler, J. Harding & D. Hofmann (ed.) *The Oxford handbook of Neolithic Europe*: 711–28. Oxford: Oxford University Press.
- Vandkilde, H. 2016. Bronzization: The Bronze Age as Pre-Modern Globalization. *Praehistorische Zeitschrift* 91: 103–23. <https://doi.org/10.1515/pz-2016-0005>. <https://doi.org/doi:10.1515/pz-2016-0005>.
- . 2019. Bronze Age Beginnings – a Scalar View from the Global Outskirts. *Proceedings of the Prehistoric Society* 85: 1–27. <https://www.cambridge.org/core/journals/proceedings-of-the-prehistoric-society/article/bronze-age-beginnings-a-scalar-view-from-the-global-outskirts/34E9A3EFA006D973825CABC30B93A2A0>. <https://doi.org/10.1017/ppr.2019.7>.

Reviewer #3 (Remarks to the Author):

The authors provide an impressive study of kinship, and social and cultural interactions in Early Bronze Age Austria. The archaeogenetic study is based on 138 new individuals associated with the Únêitice and Unterwöbling cultures, which revealed clear differences between the two groups regarding sociocultural practices and plausible trade networks.

I welcome the local/regional approach and their argument that patterns on interregional scale can only be explained, understood, and questioned, through in-depth analyses of smaller societal units. The author clearly shows that the microscale approach provides evidence of complexity which is not detectable through macroscale analyses. Furthermore, the authors provide a nuanced vocabulary regarding both discussions about identity and about possible differences between biological and social kinship, and contest through the results the ruling idea of a general patrilocal Bronze Age society. The scientific approach provides support for variance in sociocultural practices between societies also within the same cultural tradition. These efforts show that it is of utter importance to continue archaeogenetic research on local and regional scales to make sense of as well as nuance the narratives provided by macroscale approaches.

I argue that this is a very important study for broadening our understanding of Bronze Age social and cultural networks in Austria in particular, but on European level in general. The authors show that geographical barriers, such as the Danube River, can function as separation between populations, but also the opposite. While there were clear differences in ancestry between populations north and south of the Danube, there are clear links in ancestry following the river. This supports the river's function as an important trade route. Furthermore, the results also clearly point out that there are

local or regional variance in marital practices, which are only visible through these in-depth analyses.

The study will have significant impact to the field as it both show how geographical boundaries can function as barriers and possibilities to connect as well as, very welcomingly, nuances the present picture of strict patrilocal communities. This assumption has been contested within traditional archaeology but can now also be shown with the aid of archaeogenetics. The paper is significant since it encourages further bottom-up studies of local or regional scales, and clearly shows the advantages with this kind of approach. By conducting similar studies on new sites in other regions it is possible to add on and gain better understanding of prehistoric sociodynamics and its link to e.g. trade networks. As the authors also state, future studies should preferably also combine the archaeogenetic data with strontium isotope analysis. Laser ablation sampling might provide high resolution data of individual mobility that can deepen the understanding of both sociocultural practices such as exogamy as well as of trade networks and social status.

The material and amount of new data is impressive, and the applied methods are, to my knowledge not being a geneticist, thoroughly described and well adapted to the research questions. The use of IBD analysis opens up for high-resolution understanding of ancient migration, mobility, and sociocultural practices, and has been crucial for gaining the results provided in the paper. The conclusions of the paper are well-founded based on the aims of the paper as well as the presented results.

I think that the article provides important results as well as a nuanced and much needed discussions about their findings as well as links to previously published results on human health from the sites studied. I believe that this article will be a significant contribution to the research community and highly recommend publication! However, I have some minor suggestions that would improve the paper as well as some very minor edits that should be addressed.

Comments and suggested improvements:

Starting around line 78 and continuing to line 102: Description regarding burial practices denotes that males and females were buried in similar, or different manners. Are all individuals in these graves sexed through osteological and/or chemical analyses or are the authors referring to archaeological or mixed gender/sex determination of the graves? If so, I suggest that the terminology should be changed to men and women to acknowledge the distinction between social and biological sex. In line 100-102 the authors change the terminology to men and women and also point out potential complexities regarding gender identity.

Since those paragraphs were based on archaeological or mixed gender/sex determination we followed the reviewers suggestion and changed the terminology to men and women.

Lines 126-127: Dataset of 138 new individuals – it would be helpful to get information about the relation in samples between the two study areas, maybe in brackets.

We changed the sentence to this: As a result, we present a newly reported dataset comprising 138 individuals from the Early Bronze Age in Lower Austria (32 South of the Danube and 106 North of the Danube).

Lines 162-166: The authors inform about the presence of LP in the individuals; although the information is important and interesting, the statement reads a little abrupt and perhaps a little out of place since research on the frequency of LP in these individuals have not been introduced earlier in the paper. I suggest introducing this question in the paragraph spanning lines 111-115 or omitting

this part since it is not later followed up.

We omitted this part in the main text and moved it to the SI material.

Lines 201-209, especially line 207: Here, the number of shared links in male/females are provided. Both cultures exhibit higher male averages. However, the authors state that the Unterwölbling culture conversely have higher male than female averages. I am not sure about how this is conversely, since both cultures seem to exhibit higher link values in males. Either there seems to be a case of the use of wrong word, or, the text needs to be more clearly written.

Yes, this was not phrased precise we changed the sentence now to: Conversely, in the Unterwölbling culture the difference between female and male is notably smaller, females average 5 links compared to 5.6 in males.

Lines 236-242: Here the authors state that there are two instances where adult females have been buried at the same burial ground as siblings, parents, and offspring, hence showing exceptions to female exogamy. The authors also provide a description of typical burial practices in the beginning of the paper, where Únětice burials are not gendered. Could there be a possible link between these ungendered burial practices and the less strict marital practices that could be explored in the discussion? Are there local/regional differences in burial practice?

Absolutely, this is just what we are arguing. We added a line to emphasize this point in the discussion.

Minor edits:

Line 71: I am not convinced about the use of “archaeological treasures”; I think the wording sounds unscientific and a little colonial. I suggest a change of word.

We changed the wording to rich archaeologic record.

Lines 75-76: “A few dozen” and “sometimes” are very unprecise values – I think that being more concrete would improve the text as it in current state makes the reader wonder about numbers.

We changed the phrasing to “cemeteries with few graves near the settlements” and “some graves are arranged in rows in small plots”. We avoid precise numbers here as this is just intended to give a flavor of the archaeological record, it is not part of the analysis.

Line 164: A full stop or ; after “...was generally low”...

We added a full stop.

Line 196: remove “s” in individuals.

Yes, we removed the s.

Lines 198-200: These sentences read oddly and needs to be revised; The sentence starting with “One adolescent boy” is incomplete.

We changed the paragraph to this: Only one subadult individual from the Unterwölbling sites Franzhausen (FZH002) displays IBD segments shared with individuals from the Únětice group (Fig. 4, separate networks for males and females in SI note 5, SI table 5), albeit beyond the 8th degree of kinship. The individual is an adolescent boy from the triple burial of Franzhausen. He was buried with an adult man and a second adolescent boy who are identified as father and son, to which he is not genetically related.

Line 313: An "l" is missing in "child"

We added the l.

Line 341: An "r" is missing in "Harvard"

We added the r.

Line 374: Identical by descent should be changed to Identity by descent.

We changed it to identity.